# Balance of activity during a critical period tunes a developing network

Iain Hunter[1][†], Bramwell Coulson[1][†], Tom Pettini[2], Jacob J Davies[1], Jill Parkin[1], Matthias Landgraf[2]*, Richard A Baines[1]*

[1]Division of Neuroscience, School of Biological Sciences, Faculty of Biology, Medicine and Health,University of Manchester, Manchester, United Kingdom; [2]Department of Zoology, University of Cambridge, Cambridge, United Kingdom

**\*For correspondence:**
ml10006@cam.ac.uk (ML);
Richard.Baines@manchester.ac.uk (RAB)

[†]These authors contributed equally to this work

**Competing interest:** The authors declare that no competing interests exist.

**Abstract** Developing neural circuits are influenced by activity and are especially sensitive to changes in activity during critical periods (CPs) of development. Changes occurring during a CP often become 'locked in' so that they affect the mature network. Indeed, several neurodevelopmental disorders have been linked to excessive activity during such periods. It is, therefore, important to identify those aspects of neural circuit development that are influenced by neural activity during a CP. In this study, we take advantage of the genetic tractability of *Drosophila* to show that activity perturbation during an embryonic CP permanently alters properties of the locomotor circuit. Specific changes we identify include increased synchronicity of motoneuron activity and greater strengthening of excitatory over inhibitory synaptic drive to motoneurons. These changes are sufficient to reduce network robustness, evidenced by increased sensitivity to induced seizure. We also show that we can rescue these changes when increased activity is mitigated by inhibition provided by mechanosensory neurons. Similarly, we demonstrate a dose-dependent relationship between inhibition experienced during the CP and the extent to which it is possible to rescue the hyperexcitable phenotype characteristic of the *para*[bss] mutation. This suggests that developing circuits must be exposed to a properly balanced sum of excitation and inhibition during the CP to achieve normal mature network function. Our results, therefore, provide novel insight into how activity during a CP shapes specific elements of a circuit, and how activity during this period is integrated to tune neural circuits to the environment in which they will likely function.

## eLife assessment

This **valuable** study combines electrophysiology and neuroanatomy with pharmacological and optogenetic manipulation in the *Drosophila* genetic model system to pinpoint the neural substrate that is influenced by altered activity during a critical period (CP) of larval locomotor circuit development. Increasing activity during the CP causes permanent network changes, manifesting in increased recovery times from seizures and altered intersegmental coordination during locomotion, thus indicating that a setpoint of network excitability is determined during the CP. Next, **compelling** experiments demonstrate that this goes along with increased excitation/inhibition ratios to single identified motoneurons and most importantly for excitability setpoint determination during the CP excitatory and inhibitory inputs are integrated such that the effect of CP hyperexcitation is rescued by the stimulation of endogenous inhibitory inputs to the motoneurons. This provides novel insight into how developing neural network excitability is tuned and how it can be entrained during the CP.

## Introduction

A developing nervous system is exposed to multiple, and unpredictable, influences that can alter mature circuit function. These include extrinsic and intrinsic factors, such as temperature, light, mechanical stress, and/or genetic mutations. Multiple studies have revealed that developing neural networks are most sensitive to neuronal activity during critical periods (CPs) – defined periods in developmental time during which plasticity peaks (*Hensch, 2005*; *Reh et al., 2020*; *Wong-Riley, 2021*). Indeed, work in several model organisms has shown that activity perturbation during a CP often becomes 'locked in' so that once the CP is closed the mature nervous system remains changed. This was first demonstrated by *Hubel and Wiesel, 1970*, who showed that blocking visual input in kittens, but not in adult cats, caused significant, lasting change in ocular dominance. Thus, visual activity during early life is necessary for normal development of vision. Since these pioneering experiments, CPs have been identified in other networks and across species, which indicates that they are a common phenomenon in the development of neural circuits (*Marley and Baines, 2011a*; *Giachello and Baines, 2015*; *Golovin et al., 2019*; *Styr et al., 2019*; *Hensch and Bilimoria, 2012*).

Recent studies posit that aberrant activity during CPs contributes to, and may even cause, neurodevelopmental disorders, including autism (*LeBlanc and Fagiolini, 2011*), schizophrenia (*Selemon and Zecevic, 2015*), and epilepsy (*Lybrand et al., 2021*). Specifically, this growing body of work suggests that excessive activity experienced during a CP may lead to increased excitability in a mature network. Research in several model organisms, including mice and *Drosophila*, supports this view by showing that mature neurons are more excitable when they experience greater activity during development (*Marley and Baines, 2011a*; *Giachello et al., 2021*; *Styr et al., 2019*). Despite this progress, however, the specifics of how activity experienced during a CP influences a nervous system remains unclear. Key questions that remain unanswered include the following: 'How does activity perturbation affect the relative contribution of excitatory and inhibitory neurons within a network?' 'Is it the extremes, or the sum of activity, experienced during a CP that is critical for mature circuit function?' It is difficult to answer these questions when conducting experiments in complex mammalian sensory networks, such as the visual and auditory networks, which have been extensively used to explore CPs (e.g. *Hooks and Chen, 2020*; *Nakamura et al., 2020*). In this respect, the *Drosophila* larval network may provide a powerful alternative.

The larval nervous system is an experimentally tractable and numerically simple model that is very well suited to studying CPs (*Hunter et al., 2021*). Larvae are amenable to a range of different assays to demonstrate the impact of manipulating activity during development on a mature nervous system. For example, the electroshock assay enables assessment of 'network robustness' by measuring the ability of the nervous system to withstand or recover from strong electrical stimulation (*Marley and Baines, 2011a*). Similarly, functional imaging can be used to visualise locomotor network dynamics (*Streit et al., 2016*). The *Drosophila* larval connectome project (which is near completion) has led to the mapping of individual neurons, neural circuits, and development of Gal4 lines (genetic driver lines) that enable cellular-level precision in experimental manipulation of the nervous system (reviewed in *Hunter et al., 2021*). It is also possible to perform electrophysiology on certain cells, such as the anterior corner cell motoneuron (aCC, a.k.a MN-1b; *Baines and Bate, 1998*; *Marley and Baines, 2011b*). Together, these techniques and tools can be applied to study and attempt to answer questions including those posed above by probing processes associated with an embryonic locomotor CP that was first described in previous work by our group (*Giachello and Baines, 2015*).

Our earlier work showed that manipulating activity during 17–19 hr after egg laying (AEL) is sufficient to induce significant and permanent change to locomotor network stability. It also established that the range of manipulation that causes this change is broad; it extends from increased excitation during this CP being sufficient to induce a seizure-like phenotype in mature larvae subjected to electroshock, to a rescue of a genetic seizure mutation by reduction of network activity during the same period (*Giachello and Baines, 2015*). However, the effects of activity perturbation on the specific cellular components of the locomotor network have not been identified. Here, we make use of the output of the connectome project to focus attention on excitatory cholinergic A18a (*Hasegawa et al., 2016*) and inhibitory GABAergic A31k (*Schneider-Mizell et al., 2016*) interneurons, as well as the aCC motoneuron which these interneurons are monosynaptically connected to (*Giachello et al., 2022*). Specifically, we perform patch-clamp experiments on aCC to show how synaptic connectivity of these single-cell types (A18a or A31k) is influenced following activity perturbation during the embryonic

CP. We show that excessive activity during the CP permanently increases the strength of the input of A18a relative to A31k, which is consistent with a significant change in the excitation:inhibition (E:I) balance of the network. Next, we show that restoring E:I balance rescues the network, thus providing evidence that the CP integrates the 'averaged' activity that occurs while it is open. Our data suggest that 'average' activity must be balanced, so that the intrinsic properties that dictate the function of the mature locomotor network are set within normal ranges.

## Results

### Increasing neural activity during embryogenesis permanently alters network function and synchronicity

To manipulate neural activity in developing embryos, we fed gravid (i.e. egg-laying) female flies picrotoxin (PTX), which increases neural activity by non-competitive antagonism of the $GABA_A$ receptor (*Lin et al., 2012*). This procedure is sufficient to expose embryos and the resulting young larvae to PTX. Larval metabolism and excretion during early to mid-developmental stages removes PTX, so that it is no longer detectable when assaying for effect in wall-climbing third-instar larvae (L3; *Marley and Baines, 2011a*). Crucially, our previous work has shown that permanent changes in network function observed as a result of embryonic exposure to PTX occur due to the change in activity during the CP, which lasts from 17 to 19 hr AEL (*Giachello and Baines, 2015*). Similar activity manipulation during larval stages is without effect (Sarah Doran and RAB, unpublished data). We measured

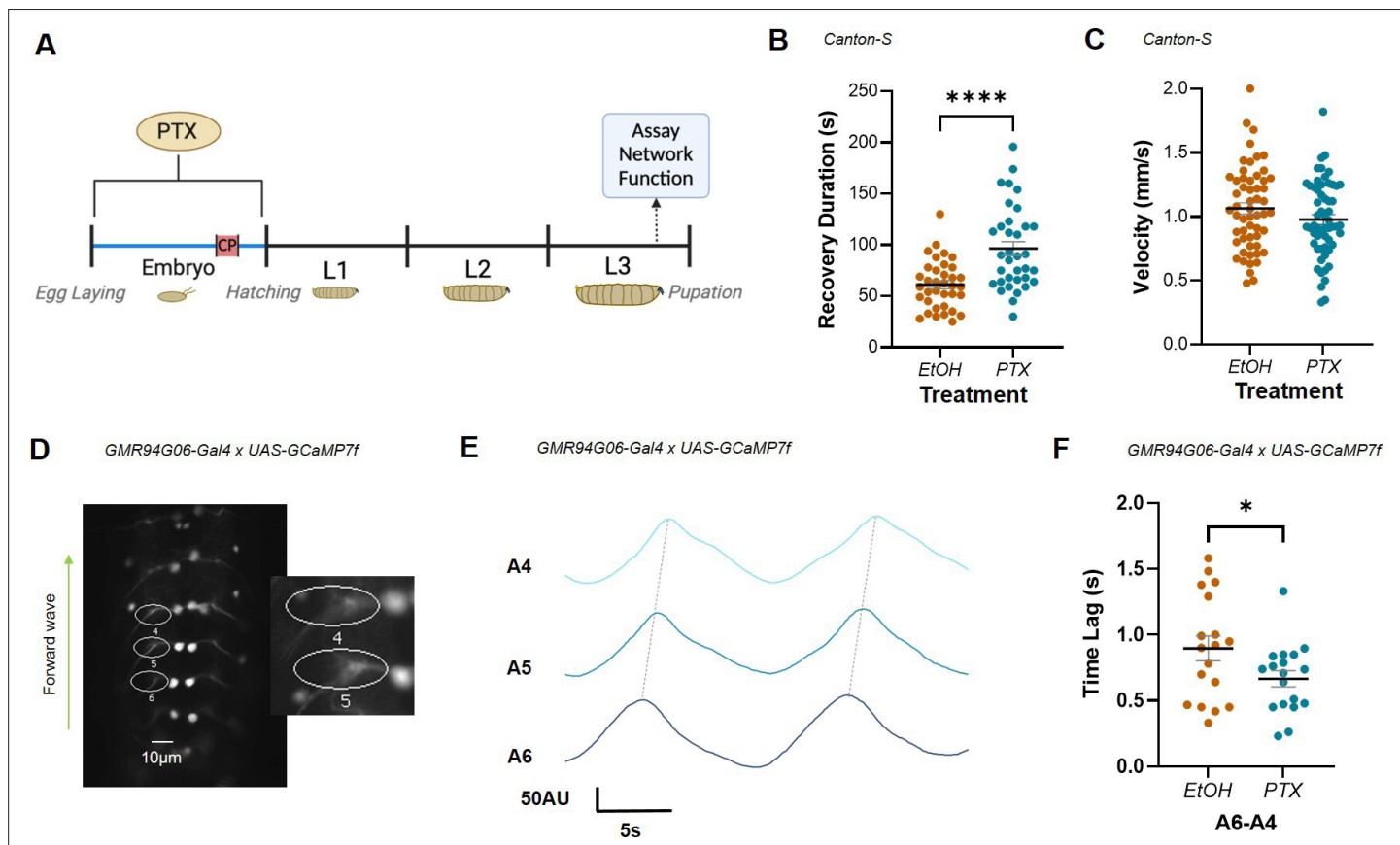

**Figure 1.** Increased excitation during the embryonic critical period (CP) alters network stability and motoneuron network synchrony. (**A**) Schematic demonstrating timeline for picrotoxin (PTX) exposure and assay of network function. PTX-induced increase in activity during development increases third-instar larval duration of recovery from induced seizure (**B**) but does not change crawling velocity (**C**). (**D**) ROIs captured GCaMP signal generated by forward waves passing through ipsilateral anterior corner cell (aCC) motoneuron axons in the L3 ventral nerve cord, abdominal segments (A6-4). Inset: expanded view of ROIs 4 and 5. (**E**) Representative traces recorded in three adjacent abdominal segments (A6-A4), during two forward waves. Colours represent different segments. Synchrony was measured as the time lag (s) of activity passing between adjacent segments (i.e. between peaks, dotted lines). (**F**) Embryonic exposure to PTX caused a significant increase in synchronicity across segments A6-A4 vs. EtOH controls.

the effect of PTX-linked activity manipulation during the CP on two independent parameters at L3: recovery from electroshock (which is longer in other models of CP activity manipulation [*Giachello and Baines, 2015*]) and crawling velocity (*Figure 1A–C*). Videos of seizure activity can be viewed in *Marley and Baines, 2011b*. Embryonic exposure to PTX significantly increased the duration of electroshock-induced seizures compared to controls (61.11 ± 3.94 s, EtOH; 96.49 ± 6.81 s, PTX; n = 36, 35; p<0.0001) (*Figure 1B*). PTX exposure did not, however, influence crawling velocity (1.06 ± 0.04 mm/s, EtOH; 0.98 ± 0.04 mm/s, PTX; n = 56, 58; p=0.14) (*Figure 1C*). This suggests that changes caused by PTX exposure during the CP were not significant enough to alter circuit function under 'normal' conditions (i.e. crawling) but clearly altered network robustness: the ability of the network to respond to strong stimulation (electroshock).

With a network-level phenotype for perturbation of CP activity established (increased seizure), we tested for possible correlates at the level of circuit activity within the locomotor network. In larvae, normal locomotor function is characterised by stereotyped activity that is regulated by a central pattern generator (CPG) (*Pulver et al., 2015*). This pattern is formed by activity waves that pass between successive segments in the ventral nerve cord (VNC) to support peristaltic locomotion. The timing (or synchronicity) of waves crossing adjacent segments is characteristic and can be measured as the time lag between peaks of activity observed in adjacent aCC motoneurons. Previous work has shown that synchronicity increases (i.e. time lag reduces) in the presence of mutations that predispose animals to seizure (e.g. *para*[bss]) or following acute exposure to proconvulsants (e.g. 4-aminopyridine or PTX; *Streit et al., 2016*). We therefore quantified synchronicity of activity between aCC motoneurons at L3 to investigate how embryonic exposure to PTX alters activity in the mature locomotor network (*Figure 1D–F*).

Increasing neural activity during embryogenesis (PTX) significantly increased synchronicity across segments A6-A4 compared to vehicle control (0.9 ± 0.09 s, EtOH; 0.67 ± 0.06 s, PTX; n = 18, 18; p=0.05) (*Figure 1F*). This increased synchronicity of activity between adjacent segments is consistent with, and may contribute to, the seizure phenotype observed. It likely reflects significant changes in CPG properties that make it harder for larvae to recover from electroshock.

## Increased activity during the CP alters the E:I balance within the locomotor network

Motoneuron activity reflects the integration of the excitatory and inhibitory premotor synaptic drive to which the cell is exposed. We therefore tested the effect of embryonic exposure to PTX on the synaptic drive provided by two identified premotor interneurons – one exciter and one inhibitor – which are monosynaptically connected to aCC (*Giachello et al., 2022*). Specifically, we quantified the strength of cholinergic exciter A18a (a.k.a. CLI2; *Hasegawa et al., 2016*) and GABAergic inhibitor A31k (*Schneider-Mizell et al., 2016*) to aCC at L3, following PTX-induced activity perturbation during embryonic development. To achieve this, premotor interneurons were activated using optogenetics, while whole-cell patch recordings were made from aCC (*Figure 2A*, left).

Larval motoneurons do not generate action potentials (APs) without synaptic drive. Thus, to determine how synaptic inputs may have changed following embryonic CP manipulation, current was injected into aCC to evoke stable firing before premotors were stimulated. Evoked firing before optogenetic stimulation, in larvae expressing A18a-Gal4, was different between control L3 (EtoH) and those exposed to PTX (6.87 ± 0.46 Hz, EtOH; 10.41 ± 1.44 Hz, PTX; n = 8, 8; p=0.03). While this difference may reflect a change in excitability due to PTX exposure, it may also have been due to the requirement for injecting differing amounts of current to maintain spiking at a constant rate. Regardless, optogenetic activation of excitatory A18a increased AP frequency in aCC in control L3 (*Figure 2A*, right), and this increase was significantly larger in L3 derived from embryos exposed to PTX (7.24 ± 2.04 Hz, EtOH; 15.81 ± 3.14 Hz, PTX; n = 8, 8; p=0.04) (*Figure 2B*). Evoked firing before optogenetic stimulation, in larvae expressing A31k-Gal4, was not significantly different between control L3 (EtOH) and those exposed to PTX (7 ± 0.94 Hz, EtOH; 7.36 ± 1 Hz, PTX; n = 8, 9; p=0.8). Activation of inhibitory A31k decreased AP firing in control L3 (*Figure 2A*); however, this decrease was not significantly different in L3 derived from embryos exposed to PTX (23.71 ± 6.76 Hz, EtOH; 28.24 ± 2.4 Hz, PTX; n = 8, 9; p=0.15) (*Figure 2C*). Comparing the magnitude of the changes in A18a or A31k input using an effect size calculation (see 'Materials and methods' for details) corroborates that excitation increased more than inhibition, as Cohen's *d* = 1.14 (large effect size) for A18a and

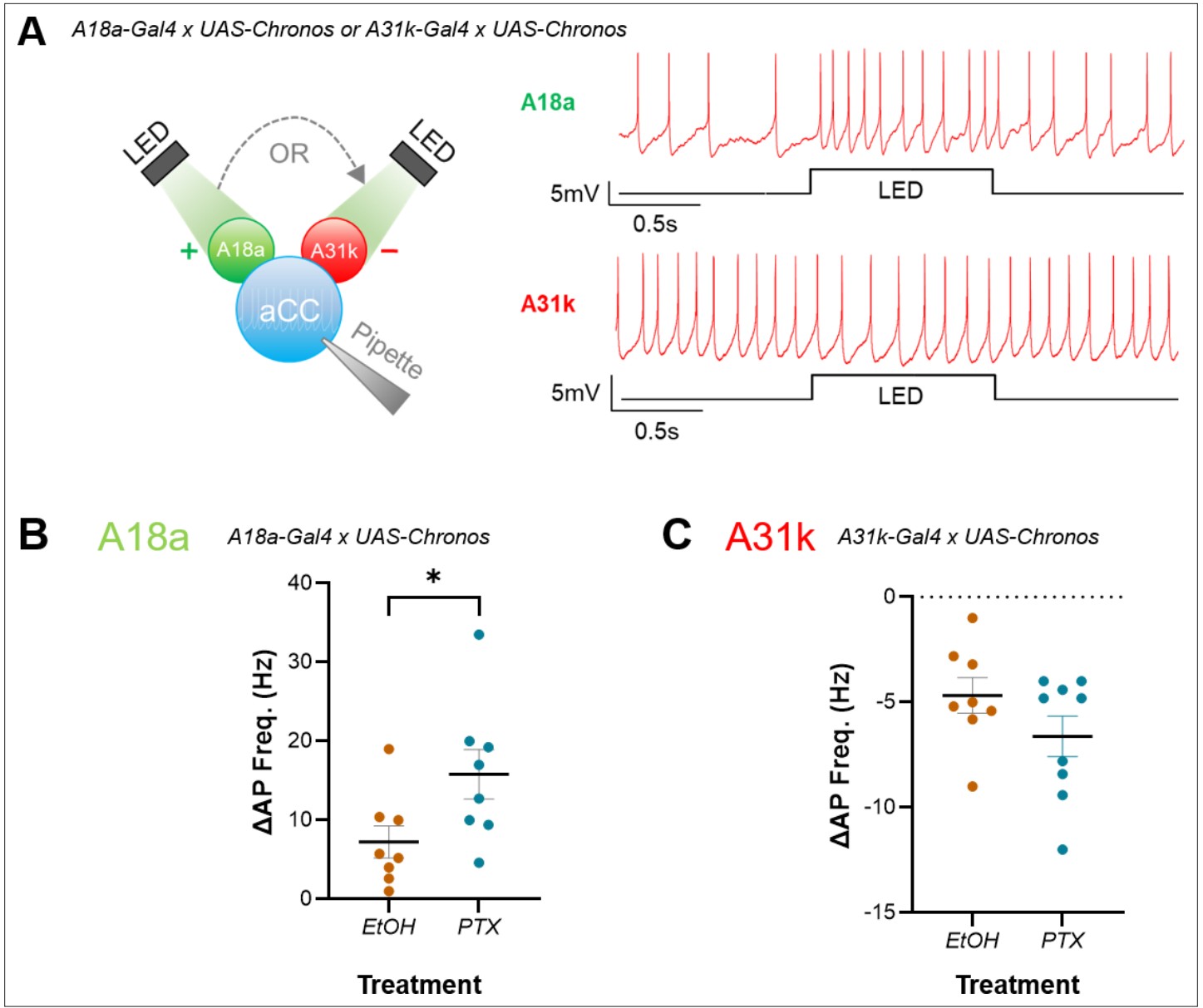

**Figure 2.** Increased excitation during the critical period (CP) preferentially increases the strength of excitatory inputs to anterior corner cell (aCC). (**A**) Left: schematic demonstrating the use of optogenetics for testing strength of A18a or A31k synaptic drive on evoked action potential firing in aCC. Right: representative traces illustrating the effect of excitatory or inhibitory input on evoked action potential firing in aCC, respectively. (**B, C**) Current-clamp recordings wherein current (~10 pA) was injected to evoke action potential firing in aCC, prior to optogenetic stimulation of premotor input. Amplitude of inputs was quantified as change in frequency of action potential firing (ΔAP Freq. (Hz)) following optogenetic stimulation of A18a or A31k. Change in action potential firing frequency in A18a (p=0.04), but not A31k (p=0.15), was significantly potentiated by embryonic exposure to PTX vs. vehicle controls (EtOH).

0.78 (medium effect size) for A31k. Measurement of $V_m$, $R_{in}$, and capacitance showed no significant changes due to the presence of PTX ($V_m$: –47.8 ± 1.26 mV, EtOH; –47.6 ± 0.78 Hz, PTX; n = 16, 17; p=0.84); ($R_{in}$: 765.4 ± 27.05 MΩ, EtOH; 746.1 ± 28.13 MΩ, PTX; n = 16, 17; p=0.62); (cap: 23.2 ± 1.0 pF, EtOH; 23.1 ± 2.6 pF, PTX; n = 16, 17; p=0.95). Our data suggests that embryonic exposure to PTX induced a shift in the E:I balance of the larval locomotor circuit to favour excitation, which is consistent with the network-level phenotypes observed (*Figure 1*). Thus, our results suggest that increasing activity during the embryonic CP increases the strength of excitatory, but not inhibitory, premotor synaptic drive.

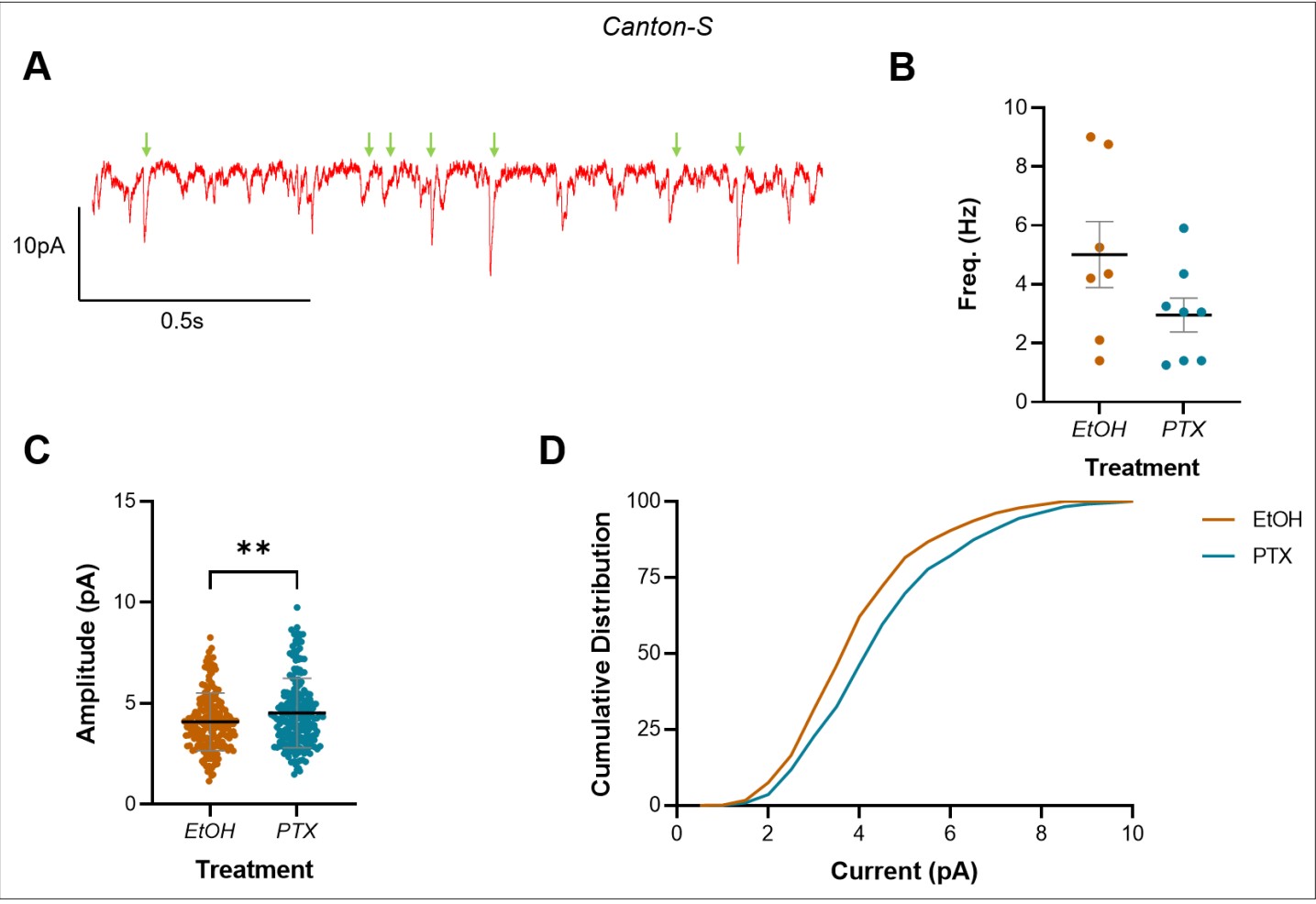

**Figure 3.** Increasing activity during the embryonic critical period (CP) increased mini amplitude but not frequency in L3 anterior corner cell (aCC).
(**A**) Representative trace showing minis (green arrows) in L3 aCC. Events not marked are not mini's and probably represent single channel events
(see ***Figure 4—figure supplement 1*** for an expanded version of this same trace). (**B**) Mini frequency, as events observed in the first 20 s of a
whole-cell voltage-clamp recording ($V_m$ held at –60 mV), show no difference between vehicle controls (EtOH) and L3 exposed to picrotoxin (PTX)
during embryogenesis. (**C**) Mini amplitude from the same recordings reveals significantly larger events in larvae that were exposed to PTX during
embryogenesis vs. vehicle controls (EtOH). (**D**) Cumulative distribution (%) of mini amplitude demonstrates a rightward shift in current size, following
activity manipulation.

## Increased excitatory input to motoneurons reflects increased postsynaptic sensitivity to quantal release

Increased excitatory synaptic drive to aCC may be due to either or both presynaptic and postsynaptic
adaptations to manipulation of activity during embryogenesis. Thus, we sought to understand which
of these effects may be responsible for the changes we observed. A change in the frequency of mini
excitatory postsynaptic potentials (mEPSPs, a.k.a. minis) would suggest the adaptation is primarily
presynaptic (e.g. increased probability of release), whilst a change in distribution and/or amplitude of
minis is more consistent with a mechanism acting postsynaptically (e.g. increased or altered receptor
subunits). We quantified the mean amplitude, cumulative distribution, and frequency of minis. Specif-
ically, we began by recording spontaneous minis in wild-type larvae following embryonic exposure to
PTX (***Figure 3A***).

Whole-cell voltage-clamp recordings, made in 2 µM tetrodotoxin (TTX, which limits synaptic release
to minis) and with $V_m$ held at –60 mV, showed that manipulating activity during embryogenesis with
PTX did not statistically change mini frequency (5.01 ± 1.12 Hz, EtOH; 2.96 ± 0.57 Hz, PTX; n = 7, 8;
p=0.11) (***Figure 3B***). In contrast, PTX treatment led to a significant increase in mini amplitude (4.08
± 1.42 pA, EtOH; 4.51 ± 1.71 pA, PTX; n = 188, 199; p=0.008) (***Figure 3C***), causing a corresponding

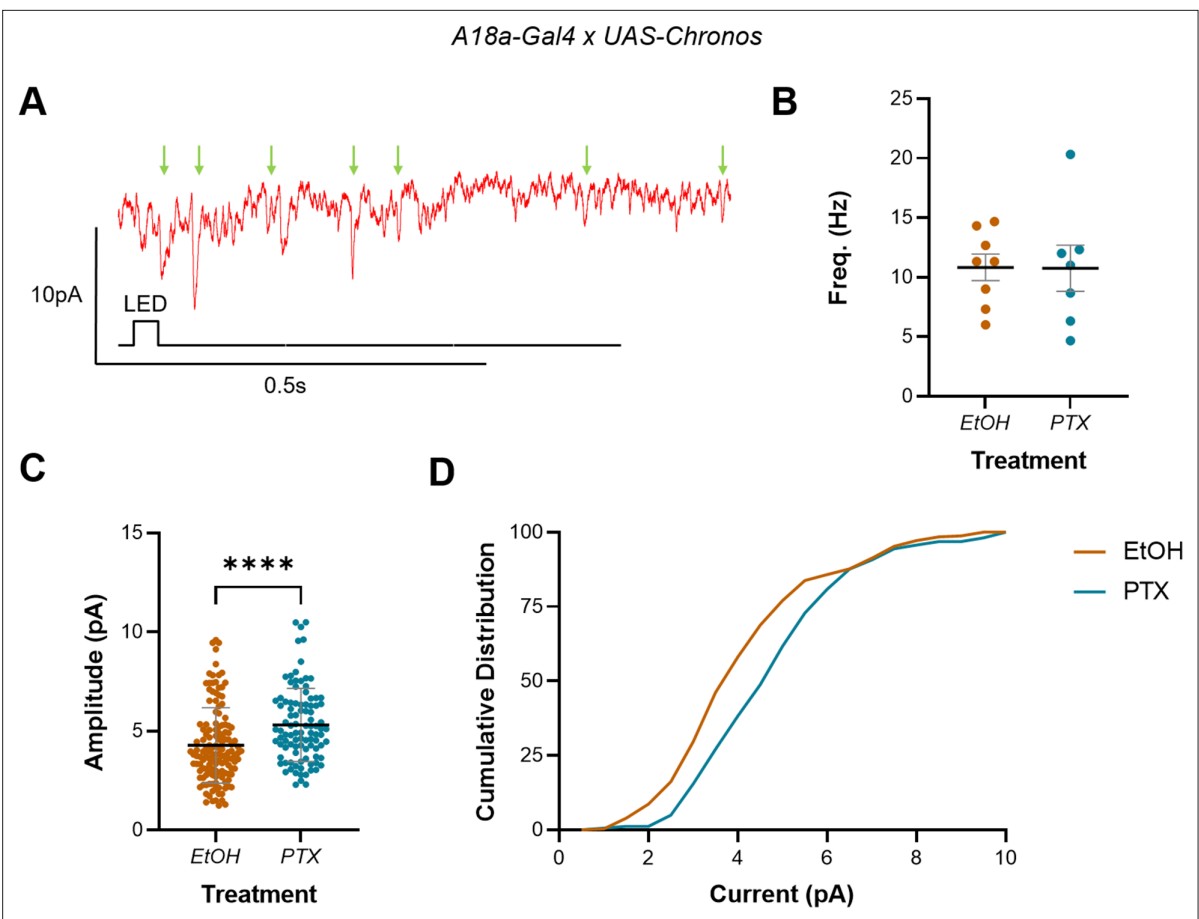

**Figure 4.** Increasing activity during the embryonic critical period (CP) increased A18a-specific mini amplitude but not frequency in L3 anterior corner cell (aCC). (**A**) Representative traces showing minis (mini excitatory postsynaptic potentials [mEPSPs], green arrows) in L3 aCC ($V_m$ held at –60 mV), following optogenetic excitation of A18a. (**B**) Mini frequency, measured as events observed in the 3 s following optogenetic stimulation of A18a (50 ms light pulse), was not different between vehicle-only controls (EtOH) and L3 exposed to picrotoxin (PTX) during development. (**C**) Mini amplitude was significantly larger following exposure to PTX during embryogenesis, vs. controls (EtOH). (**D**) Cumulative distribution (%) of mini amplitudes demonstrates a clear rightward shift in current size, following activity manipulation.

The online version of this article includes the following figure supplement(s) for figure 4:

**Figure supplement 1.** Optogenetic activation of A18a in 6 mM $Sr^{2+}$ saline stimulates asynchronous release of minis.

rightward shift in the cumulative frequency distribution (%) of currents recorded (*Figure 3D*). Thus, our recordings of minis in aCC are consistent with a change in the postsynaptic response to excitatory quantal release. However, this conclusion is based on measuring all excitatory quantal inputs to aCC, and not specifically from A18a. It was, therefore, important to refine this result by restricting it to the specific excitatory quantal input from this premotor interneuron.

It is technically challenging to record minis elicited in response to quantal release from a defined neuron because doing so requires isolating minis from a single neuron (e.g. A18a) from those released by other excitatory premotors (there are many which synapse with aCC; *Zarin et al., 2019*). To maximise our ability to do so, we adapted a technique developed in mammals, which substitutes extracellular $Sr^{2+}$ (8 mM) for $Ca^{2+}$ (2 mM) (*Xu-Friedman and Regehr, 1999*; *Bekkers and Clements, 1999*). This substitution promotes asynchronous vesicular release (i.e. occurrence of single quantal events) following firing of an evoked AP by a presynaptic neuron, which can be recorded in a postsynaptic neuron as a series of minis. Thus, we replaced $Ca^{2+}$ (2 mM) with $Sr^{2+}$ (6 mM, which better supported survival of larval neurons than 8 mM) and used optogenetics to stimulate A18a, while recording from aCC. We validated this approach by showing that optogenetic activation of A18a, in the presence of $Sr^{2+}$, greatly reduced evoked synaptic current but increased the frequency of minis recorded in aCC (*Figure 4—figure supplement 1*), which is consistent with asynchronous release. Thus, under these

conditions, stimulation of A18a produced a series of minis predominantly released from this premotor interneuron (*Figure 4A*).

Analysis of whole-cell patch-clamp recordings ($V_m$ held at –60 mV) showed that there was no change in mini frequency, in the 3 s following optogenetic stimulation of A18a, in PTX vs. controls (10.83 ± 1.12 Hz, EtOH; 10.76 ± 1.93 Hz, PTX; n = 8, 7; p=0.97) (*Figure 4B*). There was, however, a significant increase in mini amplitude (4.28 ± 1.91 pA, EtOH; 5.31 ± 1.85 pA, PTX; n = 138, 98; p<0.0001) (*Figure 4C*) and corresponding rightward shift in the cumulative distribution of current amplitudes (*Figure 4D*). These data are consistent with a change in the postsynaptic sensitivity of aCC to quantal release, which alters the E:I balance in the L3 locomotor circuit, following manipulation of activity during embryogenesis. Moreover, that we see the same effect in the presence of external $Ca^{2+}$ (*Figure 3*) likely indicates that sensitivity to all excitatory inputs has increased.

## Increased excitation during the CP does not alter motoneuron morphology or synaptic connectivity

Given our mini recordings suggested that the change in E:I balance may be associated with a change in the postsynaptic response to neurotransmitter release, we used high-resolution confocal microscopy to characterise the morphology and synaptic connectivity of aCC, following embryonic treatment with PTX (*Figure 5*). This analysis demonstrated no obvious change in dendritic arbor length, number of branch points, or terminals (*Figure 5E*). Furthermore, the number of excitatory postsynaptic sites (Drep2 puncta number) and synaptic density (μM dendrite per synapse) of aCC was unchanged following exposure to PTX during embryogenesis (*Figure 5F*). This suggests that the changes in mini amplitude, other physiology, and seizure phenotype we observed are not due to significant changes in cellular morphology. Rather, it suggests the changes may be related to an alteration in excitatory postsynaptic receptor number and/or sub-unit composition.

## Stimulating mechanosensory neurons during the CP rescues the seizure phenotype associated with excessive excitatory activity

As described above, developing circuits can be exposed to various influences (light, temperature, mutations, etc.) which may alter levels of activity. While some of these influences will increase excitation, others may decrease it. Because our data suggest that experiencing excessive excitation during the CP leads to an imbalanced mature network, it is conceivable that the nervous system must experience balance or a particular sum of excitatory and inhibitory influences, during the CP, to encode normal network properties (*Fagiolini and Hensch, 2000*).

To test this hypothesis, we combined manipulations of predicted opposite valence to see whether, in the instance that we could achieve approximately equal strengths, increased excitation and inhibition would cancel each other. We chose to stimulate chordotonal (ch) neurons to counteract PTX. This is because ch neurons are mechanosensitive cells that have been proposed to provide an inhibitory 'brake' that slows locomotor network activity during the CP (*Carreira-Rosario et al., 2021*). We, therefore, built a device to stimulate ch neurons (see 'Materials and methods' and *Figure 6—figure supplement 1*) and performed experiments in which they were activated during the CP (17–19 hr AEL) or in immediately adjacent periods (15–17 hr or 19–21 hr AEL) to assess the impact of the brake alone, before pairing it with PTX (*Figure 6A and B*).

Activation of ch neurons during the CP, but not before or after it, significantly lengthened recovery from induced seizures in L3 larvae (72.69 ± 6.02 s, 15–17 hr AEL; 122.7 ± 8.43 s, 17–19 hr AEL; 65.83 ± 26.07 s, 19–21 hr AEL; n = 39, 38, 40; p<0.0001 for 15–17 hr vs. 17–19 hr and 17–19 hr vs. 19–21 hr, 0.58 for 15–17 hr vs. 19–21 hr) (*Figure 6B*). This result shows that the majority of ch neuron activity influence on mature circuit function is, like PTX, limited to the embryonic CP. Similarly, that stimulating ch activity during the CP increases the seizure response to electroshock is consistent with optogenetic inhibition causing the same effect (*Giachello and Baines, 2015*). This result also supports previous work that showed perturbing activity of sensory neurons during the locomotor CP causes lasting changes to behaviour (*Fushiki et al., 2013*; *Carreira-Rosario et al., 2021*).

Next, we tested how ch activity impacted the phenotypes observed for recovery from electroshock, network synchronicity, and amplitude of inputs from A18a to aCC, following embryonic exposure to PTX (*Figure 6C–E*). Remarkably, ch neuron activation provided a complete rescue of both network and cellular-level phenotypes associated with embryonic exposure to PTX. Thus, recovery

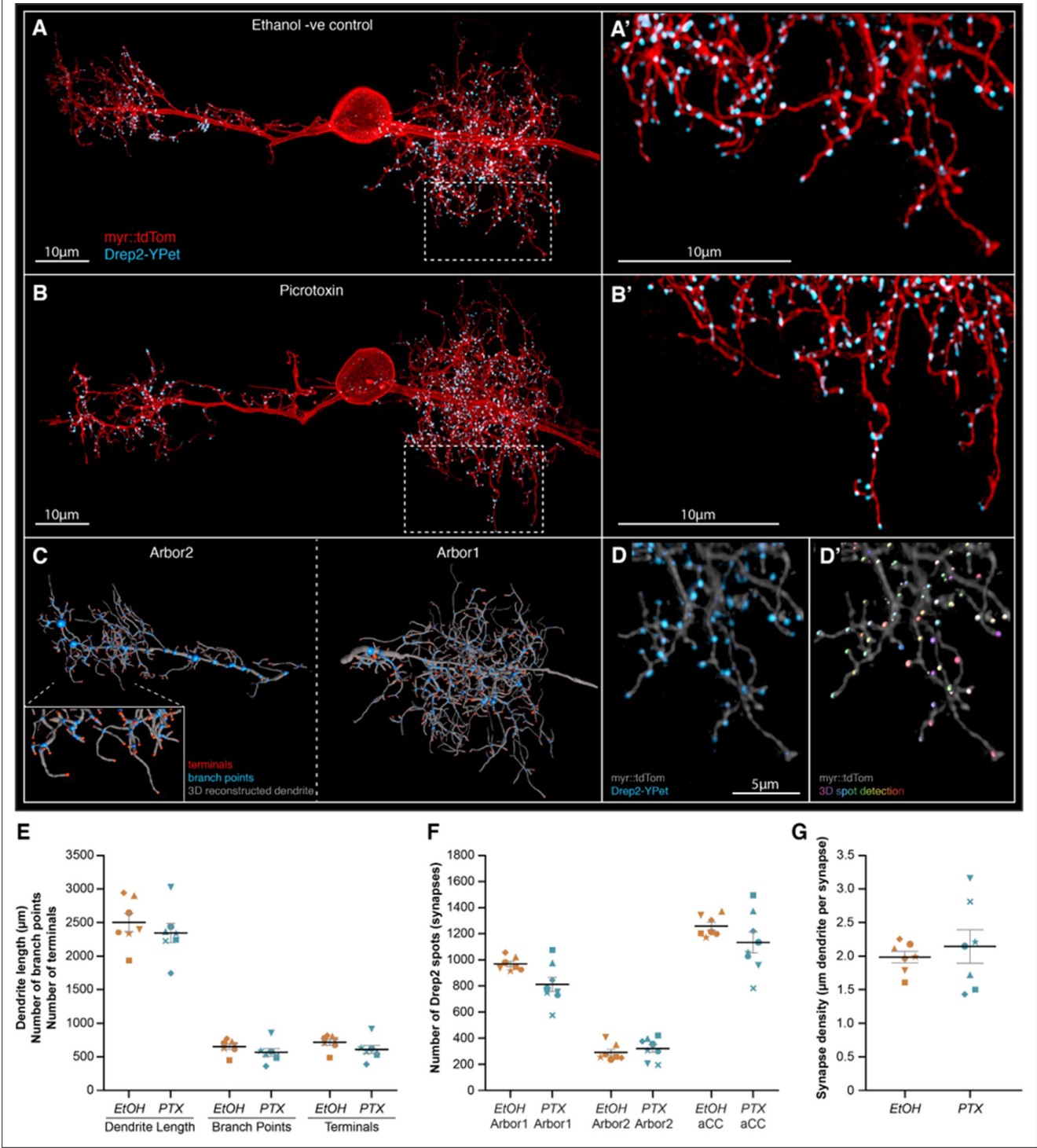

**Figure 5.** Increasing neural network activity by picrotoxin (PTX) exposure during development does not alter motoneuron dendritic arbor size or excitatory postsynaptic sites. (**A, B**) Representative images of anterior corner cell (aCC) motoneurons from larval ventral nerve cords 72 hr after larval hatching (ALH), from control (**A**) or embryonic exposure to picrotoxin (**B**). aCC dendritic arbor structure is marked by membrane targeted myr::tdTomato (red), and postsynaptic excitatory sites are marked by a Drep2::YPet fusion protein (cyan). Visualisation of individual aCC motoneurons is achieved by an RN2-enhancer element driving a stochastic FLP-out LexA expression system. (**C**) tdTomato fluorescence was used for AI-supported 3D reconstruction of the whole aCC dendritic arbor, allowing quantification of total dendrite length (grey segments), branch points (blue spots), and terminals (red spots). (**D**) Following live imaging, nerve cords were fixed and stained using a fluorophore-tagged nanobody to detect the Drep2::YPet fusion protein (**D**), enabling high-resolution and photostable imaging of Drep2::YPet puncta for quantification of excitatory postsynaptic site number

*Figure 5 continued on next page*

Figure 5 continued

by semi-automated spot detection (**D'**). (**E**) The total dendrite length, number of branch points, and number of terminals are shown for both control and picrotoxin treatments. (**F**) The number of Drep2::YPet puncta for the larger ipsilateral arbor ('arbor1') the smaller contralateral arbor ('arbor2'), and the combined sum across both arbors ('aCC') is shown for both control and picrotoxin treatments. (**G**) Synapse density, calculated as the total dendrite length divided by total Drep2::YPet positive postsynaptic sites. In (**E–G**), each point represents a single aCC, where the symbol shape denotes the particular cell.

from electroshock, which was again increased following exposure to PTX, was rescued to control levels when PTX was paired with ch neuron activation (72.03 ± 5.71 s, EtOH; 194.7 ± 12.22 s, PTX; 92.75 ± 8.95 s, PTX + Ch; n = 40, 40, 40; p<0.0001 for EtOH vs. PTX and PTX vs. PTX vs. PTX + Ch, 0.58 for EtOH vs. PTX + Ch) (*Figure 6C*). PTX-induced increase in synchrony of activity between adjacent aCC motoneurons was also rescued (0.96 ± 0.07 s, EtOH; 0.63 ± 0.06 s, PTX; 0.88 ± 0.08 s, PTX + Ch; n = 27, 28, 27; p=0.004 for EtOH vs. PTX, 0.75 for EtOH vs. PTX + Ch, 0.03 for PTX vs. PTX + Ch) (*Figure 6D*). Finally, the increased strength of excitatory A18a inputs to aCC following exposure to PTX was rescued by stimulation of ch neurons during the CP (12.92 ± 1.704 Hz, EtOH; 22.79 ± 2.42 Hz, PTX; 14.75 ± 1.98 s, PTX + Ch; n = 8, 8, 8; p=0.008 for EtOH vs PTX, 0.53 for EtOH vs. PTX + Ch, 0.023 for PTX vs. PTX + Ch) (*Figure 6E*). These results support the hypothesis that CPs integrate the excitatory and inhibitory activity to which they are exposed, to compute a sum that leads to a corresponding output in the mature locomotor network. If that sum equals balance (i.e. appropriate relative amounts of each), normal physiological function is established. An imbalance, however, may lead to altered network function.

## Balancing activity during the CP rescues adult behaviour caused by genetic mutation

Next, we sought further confirmation that CPs integrate the activity they are exposed to. Specifically, we tested the effect of different amplitudes of ch neuron activity (as a function of 0, 40 dB, 60 dB, and 80 dB speaker volume), on recovery from seizure in bang-sensitive (*para^bss^*) L3. *para^bss^* mutants express a hypermorphic variant of the sole *Drosophila* voltage-gated sodium channel (*para*), which makes neurons more excitable. Previous work has shown that excitation is increased at all stages of embryonic and larval development, resulting in increased synchrony between aCC neurons and increased duration of electroshock-induced seizures (*Marley and Baines, 2011a*; *Streit et al., 2016*; *Giachello and Baines, 2015*). These larvae, therefore, provided an alternative means of inducing excessive excitation during the embryonic CP (rather than PTX) that we could oppose by ch neuron activity to test for impact on recovery from electroshock (*Figure 7*).

In agreement with previous work (*Giachello and Baines, 2015*), *para^bss^* recovery from electroshock was significantly longer than wild-type controls (168 ± 6.96 s, bss; 87.74 ± 3.45 s, *CS*; n = 50, 50; p<0.0001). This difference was rescued completely when ch neurons were stimulated during the CP at 80 dB (87.74 ± 3.45 s, *CS*; 94.51 ± 4.59 s, *bss* + 80 dB; n = 50, 50; p=0.91). This was the maximum strength of stimulation possible, and importantly, the effect was reduced in a dose-dependent manner as the intensity of ch stimulation during the CP decreased (125.6 ± 5.82 s, bss + 60 dB; 144.3 ± 6.61 s, bss + 40 dB; n = 50, 50; p=0.03 for bss vs. bss + 40 dB, <0.0001 for bss vs. bss + 60 dB) (*Figure 7A*).

A prediction of the hypothesis that normal network function depends on a proper E:I balance during the CP is that reduced excitation during the CP would require proportionately reduced inhibition to counteract it. To test this, we raised *para^bss^* at 21°C, which reduces the duration of evoked seizure when compared to animals raised at 25°C (168 ± 6.96 s, bss 25; 113.2 ± 4.28 s, bss 21; n = 40, 50; p<0.0001). This is presumably due to reduced levels of excitation at this developmental temperature. At 21°C, stimulating ch activity during the CP using 40 dB is sufficient to completely suppress seizure behaviour (76.24 ± 5.07 s, bss 21 + 40 dB; 86.68 ± 4.2 s, *CS*; n = 40, 40; p<0.44). Notably, ch activity at 80 dB induces a seizure phenotype (130.8 ± 5.6 s, bss 21 + 80 dB; 86.68 ± 4.2 s, *CS*; n = 40, 40; p<0.0001), which is consistent with 80 dB ch-induced inhibition outweighing *para^bss^*-induced excitation at 21°C (*Figure 7B*). Combined, these results support the hypothesis that the CP integrates the excitatory and inhibitory activity to which the locomotor network is exposed using the sum (relative balance or E:I) to set key network parameters that are then 'locked in' to the mature locomotor network. Proper balance leads to normal function, while imbalance may cause dysfunction.

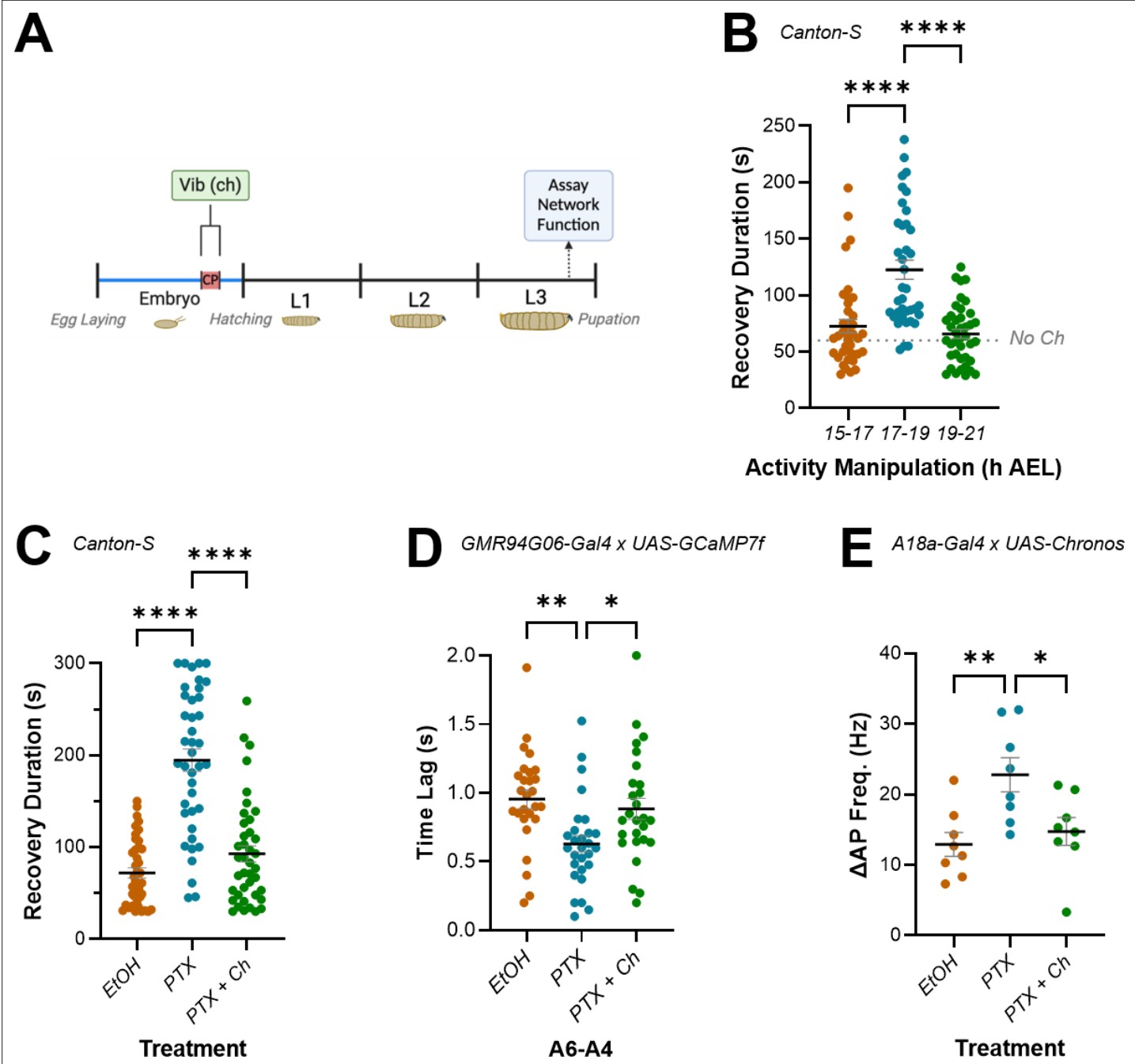

**Figure 6.** Activating mechanosensory neurons during the critical period (CP) rescues excitation:inhibition (E:I) balance due to increasing activity during embryogenesis. (**A**) Schematic showing stimulation of mechanosensory chordotonal neurons (ch) by vibration during the CP (17–19 hr after egg laying [AEL]). (**B**) Stimulating ch neurons during the CP, but not during 15–17 hr AEL or 19–21 hr AEL, significantly increases mature larval (L3) recovery duration from electroshock-induced seizure. CS is unmanipulated wild-type (Canton-S) controls. (**C–E**) Exposure to picrotoxin (PTX) during embryogenesis significantly increases mature larval (L3) recovery duration from electroshock-induced seizure, increases network synchronicity, and increases the strength of excitatory (A18a) inputs to motor neurons (aCC). All these effects are rescued by co-activation of ch neurons during the CP (PTX + Ch).

The online version of this article includes the following figure supplement(s) for figure 6:

**Figure supplement 1.** A custom-built device provides mechanical stimulation of chordotonal neurons.

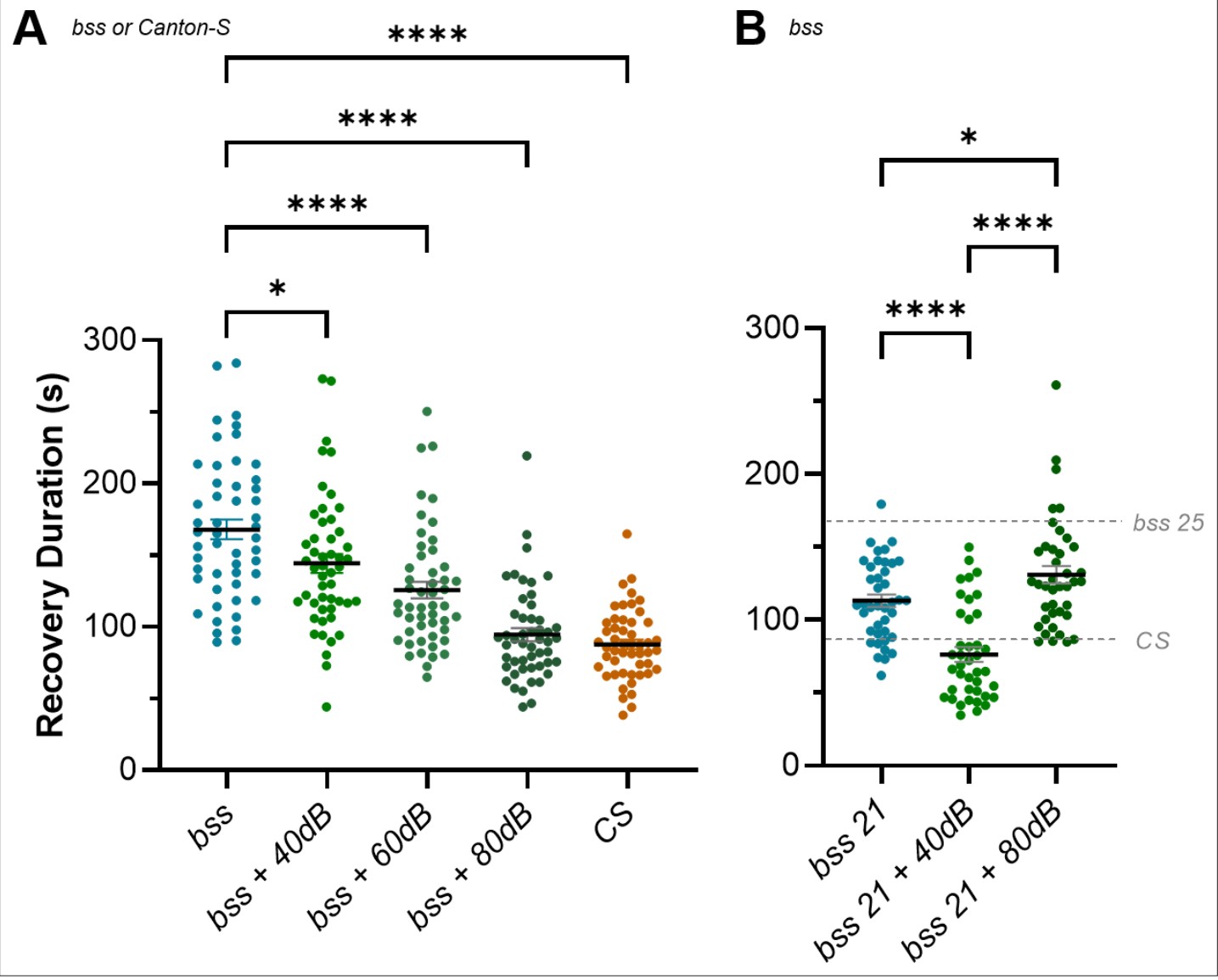

**Figure 7.** Restoring excitation:inhibition (E:I) balance during the critical period (CP) rescues seizure phenotype caused by a genetic mutation. (**A**) The *para^bss^* (*bss*) mutation causes lifelong neural hyperactivity, which is evident in a long duration of recovery from seizure when compared to wild-type controls (Canton-S). This recovery can be rescued by balancing the hyperactivity present with inhibition provided by sufficiently strong stimulation of chordotonal (ch) neurons, during the CP (*bss* + 80 dB). Weaker stimulation of the ch neuron-mediated inhibition provides intermediate phenotypes (*bss* + 60 dB, *bss* + 40 dB) that suggest the function of the mature network reflects the sum of excitatory and inhibitory activity during the CP. (**B**) *para^bss^* raised at 21°C experience less excitation than those raised at 25°C (e.g. in **A**) during development, and demonstrate a reduced seizure phenotype compared to the latter. This phenotype is completely rescued by 40 dB stimulation of ch neurons (vs. 80 dB at 25°C (**A**)), which supports the idea that it is a balanced sum of excitatory and inhibitory activity during the CP, that enables normal mature network function. Similarly, exposing *para^bss^* raised at 21°C to 80 dB vibration during the CP results in a significant increase in seizure duration. We hypothesise that at this temperature inhibition provided by ch neuron activity is greater than the increase in excitation resulting from the *para^bss^* mutation.

## Discussion

A large body of work, mostly performed in rodents, has shown that maturation of neural circuits is dependent on an interplay between excitatory and inhibitory activity, particularly during developmental CPs (*Wong-Riley, 2021*; *Hensch, 2005*; *Hensch and Bilimoria, 2012*; *Gervain et al., 2013*). CPs often reflect a change from spontaneous to patterned activity within neural networks (*Coulson et al., 2022*). Early spontaneous activity that occurs prior to CP opening is predominantly excitatory and notably includes excitatory GABAergic signalling (*Ben-Ari, 2002*). Thus, the E:I balance is strongly biased towards excitation prior to the opening of a CP (*Allène et al., 2008*). This early developmental

phase seemingly facilitates the development of neuronal electrical properties and synaptic connectivity (*Baines and Pym, 2006*). However, as sensory input becomes active, inhibitory signalling increases (with GABA now assuming its mature inhibitory signalling mode). This coincides with, and may enable, CP opening (*Hensch, 2005*; *Hensch and Bilimoria, 2012*).

Key outstanding questions related to CPs are as follows: Which network properties are set whilst a CP is open? And are these properties, which are often 'locked in', deterministic for the trajectory of a developing network? Our work provides further evidence that appropriate E:I during a CP is necessary for normal function in mature neural networks, which supports previous work (*Giachello and Baines, 2015*). Importantly, however, our work also presents novel insight into the precise role that appropriate E:I balance contributes during CPs at a level of synaptic detail that is difficult to attain in other models. We achieved this insight by exploiting the genetic tractability (courtesy of the tools developed as a consequence of the *Drosophila* connectome project; *Ohyama et al., 2015*; *Zheng et al., 2018*; *Scheffer et al., 2020*) and accessibility of the larval nervous system (reviewed in *Hunter et al., 2021*) to demonstrate that increasing activity during the CP alters mature network E:I to affect permanent change to locomotor network function.

Collectively, our results show that activity manipulation during a CP alters the E:I balance in defined components of the larval locomotor circuit. Specifically, we observe that the strength of input from an excitor (cholinergic A18a) to a motoneuron (aCC) increased more than the input from an inhibitor (GABAergic A31k). However, we recognise that other elements of the locomotor network may also undergo change due to CP manipulation. The advantage of this system is that most of these elements can be manipulated through cell-specific genetic drivers. The firing frequencies that we imposed (~5–10 Hz) are also lower than seen during fictive locomotion (*Kadas et al., 2019*), which shows burst firing lasting for ~300 ms and achieving spike frequencies of up to 100 Hz. Nevertheless, we identified an increase in the amplitude of minis which suggests the primary mechanism that alters excitability is postsynaptic. The precise mechanism contributing to increased mini amplitude remains to be determined, but a plausible scenario may involve change in cholinergic subunit composition. Similar changes to synaptic physiology have been reported in mammalian systems. For example, monocular deprivation in rat increases excitability in layer 4 of the visual cortex skewing the E:I balance towards excitation (*Maffei et al., 2004*). However, the complexity of the visual system obscures definitive conclusions because of apparent confounding observations of increasing inhibition in different components of the same circuitry (*Maffei et al., 2004*). Interestingly, these observations mirror our results where we report increases in both excitation and inhibition, but show that the former predominates, thus altering E:I balance. Moreover, and again similar to our findings, monocular deprivation in rats is sufficient to increase mini amplitude indicative of active postsynaptic mechanisms for altered excitability. Specifically, mini amplitude first declines following monocular deprivation, but after 3–6 d is potentiated due to homeostatic synaptic scaling (*Lambo and Turrigiano, 2013*). This observation suggests that homeostatic compensation can alter properties set during a CP and thus the final effect to any given network is likely to be due to a mix of CP-dependent and homeostatic-dependent change. The ability to record from identified network components and capability to explore along the entire developmental timeline of the *Drosophila* larval locomotor circuit offers the prospect of exploring this in greater detail. For example, it remains unknown if the effects we observe in late L3 are a result of only CP-related mechanisms, or are a consequence of the former plus additional homeostatic compensation occurring during larval life. The ability to record from *Drosophila* embryos (*Baines and Bate, 1998*), in addition to all larval stages, means that this important question can be addressed experimentally.

Our results show excitatory input to motoneurons is increased to a greater extent than inhibitory input when the developing circuit is exposed to excessive activity during the CP. This could be considered surprising. This is because it may be assumed that the nervous system should compensate for increased excitation by decreasing excitability through either (1) reducing excitatory (A18a) input strength; (2) increasing inhibitory (A31k) input strength; and (3) both 1 and 2. A potential explanation may be that activity during a CP is not compensated for; rather it is used to encode homeostatic set points to which the mature nervous system will continually attempt to return. Set points are defined as 'ranges of activity' associated with normal circuit function, so that activity outside of the set point may induce abnormal behaviour (*Styr et al., 2019*; *Truszkowski and Aizenman, 2015*; *Tien and Kerschensteiner, 2018*). Inappropriate E:I balance during the CP may establish a non-physiologically

relevant set point which then serves as the basis for homeostasis once the CP is closed. If CPs capture, rather than compensate for activity to establish set points, it is logical that the strength of premotor inputs increases in accordance with the hyperexcitability induced by exposure to PTX during the CP. It may also explain how the otherwise lifelong hyperexcitation caused by the *para^bss* mutation can be rescued by inhibiting neural activity during the short 2 hr CP window (*Giachello and Baines, 2015*). Alternatively, whilst the increased inhibition we observe is not statistically significant (p=0.15), it is close and has a medium effect size (Cohen's $d$ = 0.78), and thus may be indicative of an attempt by the locomotor network to rebalance activity back towards a genetically predetermined level. In this regard, it may just not have sufficient range to be able to counter the increase in excitation due to CP manipulation.

How neurons measure levels of activity during the CP remains under active investigation (*Zeng et al., 2021*; *Carreira-Rosario et al., 2021*; *Giachello et al., 2021*; *Coulson et al., 2022*; *Reh et al., 2020*; *Peters and Naneix, 2022*; *Gibel-Russo et al., 2022*). The second messenger calcium is likely to be involved as calcium signalling is central to activity-regulated plasticity during nervous system development, including adjustment to changing internal and external environments, as well as learning and memory (*Spitzer, 2012*). Similarly, we expect reactive oxygen species to have an important role. Reactive oxygen species have been identified as obligate signals required for activity-regulated synaptic plasticity (*Sobrido-Cameán et al., 2022*; *Oswald et al., 2018*; *Peng et al., 2019*; *Milton et al., 2011*; *Doser and Hoerndli, 2021*). At the same time, dysregulation in reactive oxygen species levels or buffering has been identified as one of the earliest indicators in human CP-associated neurodevelopmental conditions (*Steullet et al., 2017*). There are numerous intersections between calcium and reactive oxygen species signalling. For example, mitochondria are well-known regulators as well as sensors of intracellular calcium (*Rizzuto et al., 2012*) and a recent study has explored how in neurons activity-dependent changes in mitochondrial calcium regulate changes in metabolism, which invariably link to changes in the rate of reactive oxygen species generation (*Díaz-García et al., 2021*). We had previously shown that nitric oxide plays an important role during the CP of the *Drosophila* embryo, potentially as an activity-regulated signal that mediates adjustment within an interconnected network (*Giachello et al., 2021*). Nitric oxide might also intersect with calcium signalling as a modifier of CamKII (*Araki et al., 2020*). It will be interesting to explore both calcium and reactive oxygen species signalling further in the context of neuronal plasticity and CPs.

Our observations of lasting change to synaptic physiology provides further evidence to support the hypothesis that physiologically balanced E:I during a CP is key to normal network function. Similarly, our results that show that a ch neuron inhibitory 'brake' (*Carreira-Rosario et al., 2021*) provides a dose-dependent rescue of excessive excitation caused by PTX or the *para^bss* mutation are consistent with a CP integrating activity to which it is exposed. It is notable that 80 dB-induced ch activity rescued both the PTX and *para^bss* seizure phenotype completely. The simplest explanation for this is that 80 dB inhibition matches the strength of both types of increased excitation so they cancel each other out. That is to say, the degree of hyperexcitation caused by PTX and *para^bss* are similar. However, this may be an overly simplistic assumption and instead ch neuron activity may achieve rescue via alternate means including, but not limited to, an influence on activity-dependent CPG wave frequency during the CP.

Ch neurons have been implicated in establishing CPG rhythm via connections to premotor and pacemaker neurons A27h and M, respectively (*Zeng et al., 2021*). This rhythm-setting function may be related to relaying early muscle activity and coordinating it with the output of the CNS. Muscles produce myogenic activity and twitch early in development, beginning around 15 hr AEL (*Crisp et al., 2008*; *Crisp et al., 2011*; *Baines and Bate, 1998*). Given that this is just before the locomotor circuit CP (17–19 hr AEL) (*Giachello and Baines, 2015*), it is possible that the development of coordinated activity in the locomotor circuit depends on a succession of set points established in muscles, then peripheral neurons (e.g. ch neurons), central neurons (A18a, A31k, M, and A27h neurons), and motor neurons (aCC). Finally, despite the ch brake being inhibitory in the sense that it slows the CPG (*Carreira-Rosario et al., 2021*) and counteracts the excessive excitation provided by PTX or *para^bss*, electrophysiological recording from aCC shows that stimulating ch neurons (by optogenetics) is sufficient to depolarise aCC (IH & RAB, unpublished data). The idea that a depolarising input can inhibit activity in the locomotor circuit is likely explained by the contribution ch neurons make to the CPG rhythm. CPG activity causes regular significant depolarisation of aCC in the form of spontaneous rhythmic currents

(SRCs). SRCs may be followed by refractory periods that prevent further (i.e. greater) activity occurring in the network. This could be explored in future work.

The rescue of increased seizure in *para^bss* reported (**Giachello and Baines, 2015**) was achieved using a different modality to inhibit neurons (optogenetics) than we have used here (vibration). The latter is more physiological; it is known that mechanoreceptors provide activity that contributes to normal development of locomotor circuits/CPGs in flies and other animals (**Carreira-Rosario et al., 2021; Zeng et al., 2021**). Indeed, our validation experiment showed that activating proprioceptors during just the CP, in the absence of PTX, is sufficient to induce a seizure phenotype.

Vibration represents one of a growing list of approaches that show that manipulating neural activity during development can cause permanent change to network function. Other methods for manipulating activity include endogenous mutations (e.g. *para^bss*), exposure to drugs (PTX or phenytoin) (**Giachello and Baines, 2015**), genetically targeted expression of tetanus toxin (**Carreira-Rosario et al., 2021**), temperature (**Kiral et al., 2021**), and light (via optogenetics) **Giachello and Baines, 2015**). Some (e.g. temperature and vibration) may be encountered by animals developing in the wild and can be excitatory or inhibitory (e.g. altered temperature can increase or decrease neural activity during development). Thus, our work, in combination with that conducted by others, suggests that CPs may 'measure' neural activity which is influenced by various environmental and internal factors. This measurement is used to set key network parameters according to the context in which the circuit functions. In support of this, a recent study reports that synaptic number in the *Drosophila* adult eye scales inversely with developmental temperature (**Kiral et al., 2021**). Behavioural analysis indicates that this phenomenon ensures that adult movement is highly dependent on, and optimised for, developmental temperature.

In conclusion, our work provides novel insight into how activity manipulation, during a defined CP, results in mature network change. Specifically, we show a clear and lasting change to network E:I balance that leaves the locomotor network prone to seizure following a sufficiently large stimulus and relate this to cellular and synaptic-level changes in the circuit. The changes we observed may reflect a network that has encoded a non-physiological set point that, though capable of supporting normal locomotion, is not sufficient to guard against the effects of strong stimulation. Finally, we show that changes induced by excessive excitation can be compensated by inhibition provided by mechanosensory neurons in a dose-dependent manner. This supports the view that a CP acts to tune a developing network based on the influences on activity to which it is exposed.

## Materials and methods

**Key resources table**

| Reagent type (species) or resource | Designation | Source or reference | Identifiers | Additional information |
|---|---|---|---|---|
| Genetic reagent (*Drosophila melanogaster*) | *para^bss1*(*bss*) | Laboratory stocks | | Na_v channel hypermorph |
| Genetic reagent (*D. melanogaster*) | GMR94G06-Gal4 | Gift from Juan J. Pérez-Moreno **Pérez-Moreno and O'Kane, 2019** | | Expressed in aCC |
| Genetic reagent (*D. melanogaster*) | 15B07-Gal4 -Gal4 | Gift from Aref Zarin | | Expressed in A18a |
| Genetic reagent (*D. melanogaster*) | A31k-Gal4-Gal4 | Gift from Akinao Nose | | Expressed in A31k |
| Genetic reagent (*D. melanogaster*) | iav-GAL4 | Bloomington Drosophila Stock Center | BDSC #52273 | Expressed in chordotonal neurons |
| Genetic reagent (*D. melanogaster*) | UAS-Chronos | Bloomington Drosophila Stock Center | BDSC #77115 | Green-shifted channelRhodopsin |
| Genetic reagent (*D. melanogaster*) | UAS-GCaMP7f | Bloomington Drosophila Stock Center | BDSC #80906 | Calcium activity indicator |
| Genetic reagent (*D. melanogaster*) | LexAop2-Flp2… | Gift from Jan Felix Evers **Ostrovsky et al., 2019** | | lexAoperon flip-out line |

*Continued on next page*

*Continued*

| Reagent type (species) or resource | Designation | Source or reference | Identifiers | Additional information |
|---|---|---|---|---|
| Genetic reagent (*D. melanogaster*) | *13xLexAop2-IVS-myr::tdTom…* | Generated for use in the present study | | Modified lexAoperon flip-out line |
| Chemical compound, drug | Picrotoxin | Sigma-Aldrich, UK, or Merck, NJ | Sigma-Aldrich P1675 or Merck 528105 | Toxin that increases neural activity |
| Software, algorithm | Arduino IDE | Arduino | Arduino 1.8.19 | Environment used for programming Arduino Uno |
| Other | Arduino Uno Rev 3 | Arduino | | Printed circuit board with ATmega 328P microcontroller |
| Other | Adafruit STEMMA Speaker | Adafruit | Plug and Play Audio Amplifier | 1 W, 8 Ohm speaker |

## *Drosophila* rearing and stocks

*Drosophila* stocks were kept on standard corn meal medium, at 25°C. For all experiments where crosses were necessary, stocks were selected and combined as a 1:2 ratio of Gal4-line males to UAS-line virgin females.

Wild-type *Canton-S* or bang-sensitive (a.k.a. bss, *para^bss1^*) were from our lab stocks. The following lines were used for GCaMP imaging and/or electrophysiology: *w;;GMR94G06-Gal4* aCC, gift from Juan J. Pérez-Moreno (*Pérez-Moreno and O'Kane, 2019*); *w⁻; 15B07-Gal4* (A18a, a.k.a. A18a-Gal4, gift from Aref Zarin); *R20A03-p65ADzp in attP40; R93B07-ZpGDBD in attP2* (A31k, a.k.a. A31k-Gal4, gift from Akinao Nose); *P{iav-GAL4.K}3* (chordotonal neurons, BDSC #52273); *w[*]; P{y[+t7.7] w[+mC]=20XUAS-Chronos::mVenus}attP40* (BDSC #77115); *w[1118]; P{y[+t7.7] w[+mC]=20XUAS-IVS-jGCaMP7f}su(Hw)attP5* (BDSC #80906); *w-; 13xLexAop2-Flp2@attP40, Drep2FonYPet* gift from Jan Felix Evers (*Ostrovsky et al., 2019*), *13xLexAop2-IVS-myr::tdTom @ su(HW)attP5/CyO, Dfd-GMR::YFP; RN2-FLP-hopA, tub-FRT-stop-FRT-LexAVP16 (No.5)/TM6b, Sb, Dfd-GMR::YFP* (a.k.a. RN2-FLP LexA-tdTom Drep2::YPet) for microscopy.

## Manipulation of activity during development

### PTX feeding protocol for increasing neural activity during development

PTX (Sigma-Aldrich, UK) and vehicle food were made by pipetting 20 µl 25 mg/ml PTX, or an equivalent volume of vehicle (ethanol, EtOH) to 1 ml of ddH$_2$O, for a final concentration of 415 µm PTX. A 4 g yeast extract powder (Melford, UK) was stirred into the 1 ml ddH$_2$O to make 5 g yeast paste food. Approximately 1 ml of either PTX or control food was then placed on the surface of a 6 ml grape agar plate. Flies were added to a 50 ml beaker using a funnel, and the beaker was connected to a grape agar plate (with food) using a pressure-fit plastic collar to form a small laying pot. Laying pots were left at 18°C for 48 hr (over weekend) to allow flies to acclimate and eat the food. After 48 hr, pots were moved to 25°C and changed (beaker swapped and food replenished) at 10:00 and 15:00 every day for 5 d (Monday–Friday). Embryos were collected from the surface of grape agar plates using a wet paintbrush for the final 3 d (Wednesday–Friday) of this process. Following collection, embryos (of either sex) were gently placed on the surface of vials containing either (1) corn meal medium enriched with ~70 µM all-trans retinal (Sigma-Aldrich, 50 µl of 100 mM stock pipetted onto ~7 ml food and left to dry before embryos were added), a cofactor chromophore necessary for activation of channelrhodopsin, for animals to be used in optogenetic experiments; or (2) normal corn meal medium, in the case of embryos developing into larvae used in GCaMP imaging or behavioural experiments.

## Vibration protocol for activating chordotonal neurons during embryonic development

Gravid females were fed either (1) normal corn meal medium where effect of chordotonal neurons was not used to rescue effect of PTX (e.g. *Canton-S* and *para^bss^* experiments) or (2) 830 µm PTX food (as above), when chordotonal neuron activity was used to rescue effect of PTX. Male and female embryos were collected Monday–Thursday and timed precisely (from animals laying in a 4 hr window between 10:30 and 14:30) to ensure that the developmental stage at which embryos were exposed to

vibration included the 17–19 hr AEL CP. Embryos were collected onto grape agar plates and arranged dorsal side up. Plates were then placed on a custom-made perspex shelf glued to a speaker (Adafruit STEMMA Speaker – Plug and Play Audio Amplifier), in a 25°C incubator. The speaker was wired to an Arduino Uno Rev 3 printed circuit board with an ATmega328P microcontroller, which was programmed using C$^{++}$, via the Arduino IDE (Arduino 1.8.19), using code available on GitHub (*J-M-L et al., 2022*).

The code was written and refined by several users of the Arduino forum. The original forum thread, which includes the usernames of those who contributed to it, is visible here. The code instructed the Arduino to power the speakers to provide a 1 kHz tone (vibration) to the perspex shelf (so the embryos), with an on–off frequency of 0.5 Hz (on 1 s, off 1 s) for 6 hr, starting 12.75 hr after a button was pressed. Thus, once embryos were collected and placed on the shelf at 14:30, the button was pressed at 14:45, and stimulation was delivered to embryos between 03:30 and 09:30. Tone volume was 80 dB for all experiments besides those where others are stated explicitly (e.g. 40 and 60 dB).

## Crawling velocity

Male and female wall-climbing third-instar larvae (L3) were selected from vials and washed to remove food residue. They were dried with a tissue and placed in the centre of a 1% agarose gel arena surrounded by a 5 M NaCl (Sigma-Aldrich) moat. Animals were then left to acclimatise to the arena and recover from handling for 30 s. Crawling was then recorded for 120 s using a DanioVision observation chamber (Noldus, the Netherlands). Only larvae that spent 100% of recording time within the gel arena were included in analysis (the few that entered the moat were excluded). Velocity was calculated from movement of a centroid automatically identified by the tracking system, in mm/s, using the Ethovision software (Noldus, v. XT 11).

## Recovery from electroshock

Assays were conducted as described previously (*Marley and Baines, 2011a*). In brief, male and female wall-climbing L3 larvae were selected from vials and washed to remove food residue. L3 were dried with a tissue and placed on an plastic plate lid to recover from handling. Once animals had recovered and normal crawling had resumed, a handheld stimulator composed of two tungsten wires fixed to a nonconducting rod (tips of the two wires [0.1 mm diameter] were set 1–2 mm apart) was positioned and very gently pressed into the larval cuticle, perpendicular to and over the approximate position of the CNS. A direct current pulse (10 V, 1 s) generated by a DS2A Mk.II stimulator (Digitimer Ltd, UK) was delivered via the probe to the animals, which usually responded with a sustained contraction and subsequent immobility representing seizure-like behaviour. The time to resumption of normal behaviour (regular, whole-body peristaltic waves that produce forward locomotion) was recorded and defined as duration of recovery from seizure. Strong peripheral stimulation likely causes excessive and synchronous synaptic excitation within the CNS, resulting in seizure. However, the precise mechanism of this effect remains to be determined.

## Electrophysiology

Recordings were performed as described previously (*Baines and Bate, 1998*; *Marley and Baines, 2011a*) in male and female L3 larvae. In brief, the protocol was as follows: larvae were dissected in a dish in saline (135 mM NaCl [Fisher Scientific, UK], 5 mM KCl [Fisher Scientific], 4 mM MgCl$_2$·6H$_2$O [Sigma-Aldrich], 2 mM CaCl$_2$·2H$_2$O [Fisher Scientific], 5 mM TES [Sigma-Aldrich], 36 mM sucrose [Fisher Scientific], at pH 7.15) to remove the CNS. The isolated CNS preparation was then transferred to a 200 µl droplet of external saline, within which it was laid flat (dorsal side up) and glued to a Sylgard-coated cover slip on a slide. Then, 200 µM mecamylamine (Sigma-Aldrich) was added to the droplet of saline for recordings made following optogenetic activation of A31k, but was not added for those made following activation of cholinergic A18a.

Before recording, interneuron Gal4 and mVenus-tagged reporter line expression (e.g. 15B07-Gal4 × UAS-Chronos::mVenus) was checked by momentarily exposing preparations to blue light (470 nm OptoLED, Cairn Instruments, UK), while viewing them under a microscope (model BX51WI, Olympus, UK). Once expression was confirmed, aCC motoneurons present in the VNC segments that expressed the interneuron Gal4 and reporter were identified using bright-field microscopy. A 1% protease (Sigma-Aldrich) was applied to those segments to remove overlaying glia to facilitate access to aCC. Recordings were made using borosilicate glass pipettes (GC100F-10, Harvard Apparatus, USA)

that were fire polished to resistances of 10–15 MΩ and filled with intracellular saline (140 mM K1-D-gluconate [Sigma-Aldrich], 2 mM MgCl$_2$ · 6H$_2$O [Sigma-Aldrich], 2 mM EGTA [Sigma-Aldrich], 5 mM KCl [Fisher Scientific], and 20 mM HEPES [Sigma-Aldrich], at pH 7.4). Input resistance was measured in the 'whole cell' conformation, and only cells that (1) had an input resistance ≥0.5 GΩ, (2) had a membrane potential ≤–40 mV, and (3) were not firing APs at rest were used for experiments. Data was captured using a Multiclamp 700B amplifier controlled by pCLAMP (v. 11.2.1) via a analog-to-digital converter (model Digidata 1440A, Molecular Devices, CA, USA). Traces were sampled at 20 kHz and filtered online at 10 kHz.

Current-clamp recordings from aCC were made by injecting current (~5–10 pA) sufficient to evoke APs in aCC at a frequency of ~5–10 Hz. Premotor synaptic drive was elicited by optogenetic stimulation of interneurons using green light (565 nm OptoLED, Cairn Instruments) controlled by a Clampex (version 10.7) protocol comprised of three sweeps (repetitions) of 1 s LED off, 1 s on, 1 s off, for each preparation. Change in frequency (ΔAP Freq. [Hz]) was calculated as the difference in number of APs firing before and during optogenetic stimulation.

## GCaMP imaging

Imaging was performed on aCC motor neurons in w; UAS-GCamP7f x w;;GMR94GO-Gal4 male and female L3 derived from gravid females fed PTX, or vehicle control (see section 'Manipulation of activity during development'). Larvae were dissected in a dish in external saline (see section 'Electrophysiology') to remove the CNS. The isolated CNS preparation was then transferred to a droplet of saline, within which it was laid flat (dorsal side up) on a Sylgard-coated cover slip on a slide. The slide was positioned under a microscope (model BX51WI, Olympus) with a Hamamatsu ORCA-flash 4.0C114440 digital camera, and the CNS viewed using a ×20 water immersion lens. Blue light (a 470 nm OptoLED, Cairn Instruments) was used to excite GCaMP, and fluorescence was recorded for 180 s at a 10 Hz imaging rate using Winflour v4.1.3 (University of Strathclyde). During analysis, ROIs were placed on aCC axons, on the same side of the VNC, from A6-A4, and exported into Clampex (10.7). Traces were smoothed (add type and amount), with forward wave propagation being measured between adjacent ipsilateral ROIs. Time lag was measured as the mean duration between events (peak amplitudes), between segments A6-A5 and A5-A4. Means were calculated from up to five individual measures of time lag.

## Confocal microscopy
### PTX feeding protocol
PTX (Merck, NJ) was prepared to 25 mg/ml in 100% ethanol, then 40 ul of this stock diluted 50-fold by adding to 1960 ul ddH$_2$O then mixed with 8 g of yeast extract powder (Melford), while for controls the equivalent amount of EtOH only was used. Canton-S virgin females were crossed to RN2-FLP LexA-tdTom Drep2::YPet males (see above for complete genotype) and left to mate in vials at 25°C for 24 hr. Mated flies were distributed to two laying pots with apple juice agar plates, with an excess (~0.5 g) of either PTX or control yeast extract paste. Laying pots were kept at 25°C, and yeast plates changed and discarded daily for 3 d to ensure sufficient time for drug exposure. Thereafter, newly hatched larvae aged to within 1 hr of one another were picked from laying plates and transferred to new apple juice agar plates with an excess of normal yeast paste (dried baker's yeast and ddH$_2$O), and allowed to develop at 25°C for 72 hr.

### Imaging
At 72 hr after larval hatching (ALH), larvae of the correct genotype were chosen by selection against the Dfd-GMR-YFP balancer. Larval nerve cords were dissected at room temperature in Sorensen's saline, adhered to a poly-L-lysine (Merck)-coated coverglass, and imaged 'live' within a drop of Sorensen's saline on a Leica Stellaris 5 upright confocal microscope with a Leica 506148 HC APO L 63×/0.9 W U-V-I Water Immersion objective. The following imaging settings were used: format 800 × 800, 16 bit, scan speed 400 Hz, bidirectional scanning, pinhole 1 Airy Unit, z step size 0.3 um, sequential frame scanning. For Drep2::YFP detection: excitation 514 nm at 1.5% power, gain 20, emission collection window 520–540 nm, frame accumulation 3. For myr::tdTom detection: excitation 561 nm at 2% power, gain 15, emission collection window 570–700 nm, line accumulation 4. Inbuilt Leica LIGHTNING deconvolution software was used with default settings optimised for water immersion.

Following acquisition of the 'live' image, nerve cords were fixed for 15 min at room temperature in 2% PFA in Sorensen's saline, washed 3× in Sorensen's saline, blocked for 1 hr at 4°C in blocking solution (1× Sorensen's saline, 0.3% Triton-X-100, 0.25% BSA), and then incubated for 15 hr at 4°C with FluoTag-X4 anti-RFP STAR635P and FluoTag-X4 anti-GFP ATTO488 (NanoTag Biotechnologies), both diluted 1:500 in blocking solution. Stained samples were then washed 3× in Sorensen's saline at 4°C for >1 hr before imaging again. Imaging settings for nanobody-stained samples were the same as for live samples, with the following specific differences: format 1024 × 1024, z step size 0.25 um. For Drep2::YFP>anti-GFP ATTO488 detection: excitation 488 nm at 1.5% power, gain 2.5, emission collection window 498–580 nm, line averaging 3, line accumulation 3. For myr::tdTom>anti-RFP STAR635P detection: excitation 638 nm at 2% power, gain 15, emission collection window 648–830 nm, line averaging 3, line accumulation 3.

### Image analysis

Imaris v10.0 software (Bitplane) was used for image analysis. Dendrites were reconstructed from the myr::tdTom channel of 'live images' as dendrite morphology is clearer pre-fixation. The Imaris filaments module was used for dendrite reconstruction using the tree autopath algorithm, with the starting points threshold set to include just one starting point on the cell body. Seed points for segments multiscale points were set to a thinnest diameter of 0.01 um and largest diameter of 3 um. Iterative rounds of user-informed AI training were used for seed point classification until all seed points correctly coincided with dendrites, followed by iterative rounds of AI training for segment classification. A tree-build maximum gap length of 5 um was used. Drep2::YPet marking postsynaptic sites was detected from the fixed and nanobody-stained images since Drep2::YPet>anti-GFP ATTO488 showed good photostability through the z-stack compared with 'live' YPet protein, which suffered substantial photobleaching through the course of acquiring a 'live' image stack. The Imaris spots module was used for Drep2::YPet spot detection using an estimated xy diameter of 0.1 um and z diameter of 0.5 um. The spot quality threshold was informed by the frequency histogram and set to the trough point, at which any further decrease in threshold results in a sharp increase in spot number, corresponding to false-positive spots detected outside the cell from low-level background. In this way, all Drep2::YPet puncta of different sizes were detected, while avoiding false positives from low-level background.

## Statistical analysis

Data were imported into and organised in Microsoft Excel (Microsoft Corporation, WA), and GCaMP imaging activity peaks were smoothed and quantified in ClampFit before being copied into GraphPad Prism 9 (GraphPad Software) for statistical analysis. Normality was tested using the Shapiro–Wilk test, with further analysis conducted according to the nature of the distribution. Specifically, differences between two groups of normally distributed data for crawling velocity, recovery time from electroshock, change in aCC evoked AP firing frequency in response to premotor interneuron stimulation and mini amplitude or frequency, were analysed by Student's $t$-tests and are shown as mean ± SEM. Normally distributed data compared across three groups (e.g. all experiments that showed rescue of activity imbalance by ch stimulation, besides synchronicity [see below]) was analysed using one-way ANOVA with multiple comparisons. Data for rescue of synchronicity following exposure to PTX by ch stimulation were not normally distributed, so were analysed using one-way ANOVA with Kruskal–Wallis multiple comparisons test. p-Values<0.05 are considered significant, and levels of significance are represented by *p<0.05, **p<0.01, ***p<0.01.

Effect size (Cohen's $d$) for change in aCC-evoked AP firing frequency in response to premotor interneuron stimulation was calculated manually using the formula: $Cohen's d = (M_1 - M_2)/SD_{pooled}$, where $M_1$ = mean aCC AP firing frequency before activation of premotor neuron; $M_2$ = mean aCC AP firing frequency during activation of premotor neuron; and $SD_{pooled} = \sqrt{}$ .

## Acknowledgements

This work was supported by funding from a Joint Wellcome Trust investigator award to RAB and ML (grant 217099/Z/19/Z). Work on this project benefited from the Manchester Fly Facility, established through funds from the University and the Wellcome Trust (grant 087742/Z/08/Z).

# Additional information

## Funding

| Funder | Grant reference number | Author |
|---|---|---|
| Wellcome Trust | Grant 217099/Z/19/Z | Iain Hunter |

The funders had no role in study design, data collection and interpretation, or the decision to submit the work for publication. For the purpose of Open Access, the authors have applied a CC BY public copyright license to any Author Accepted Manuscript version arising from this submission.

## Author contributions

Iain Hunter, Bramwell Coulson, Conceptualization, Investigation, Methodology, Writing – original draft, Writing – review and editing; Tom Pettini, Investigation, Methodology, Writing – original draft; Jacob J Davies, Formal analysis, Investigation; Jill Parkin, Investigation; Matthias Landgraf, Richard A Baines, Conceptualization, Resources, Supervision, Funding acquisition, Writing – original draft, Project administration, Writing – review and editing

## Author ORCIDs

Iain Hunter ⬛ https://orcid.org/0000-0002-4506-9349
Matthias Landgraf ⬛ http://orcid.org/0000-0001-5142-1997
Richard A Baines ⬛ http://orcid.org/0000-0001-8571-4376

## Ethics

Because this work uses Drosophila, there are no ethical concerns to report.

Reviewer #1 (Public Review): https://doi.org/10.7554/eLife.91599.3.sa1
Reviewer #2 (Public Review): https://doi.org/10.7554/eLife.91599.3.sa2
Reviewer #3 (Public Review): https://doi.org/10.7554/eLife.91599.3.sa3
Author Response https://doi.org/10.7554/eLife.91599.3.sa4

---

# Additional files

## Supplementary files
• MDAR checklist

## Data availability

Figures depicting behavioural experiments show all data points including calculated means ± sems. Electrophysiology figures show analysed data derived from electrophysiology recordings. Recordings (made in pClamp) for specific experiments are available from RAB. We have not deposited these traces for open access because they are plentiful and, moreover, only sections of each recording are used and it is difficult to convey this information for all traces. Confocal images shown are representative and the associated data presents analysed data relating to these images. Access to further images can be requested from ML. Raw data, for specific experiments reported in this study, is freely available to individuals without restriction, on reasonable request, without charge nor requirement to submit a project proposal.

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
