## [Editor Report · eLife assessment]

This **valuable** study combines electrophysiology and neuroanatomy with pharmacological and optogenetic manipulation in the *Drosophila* genetic model system to pinpoint the neural substrate that is influenced by altered activity during a critical period (CP) of larval locomotor circuit development. Increasing activity during the CP causes permanent network changes, manifesting in increased recovery times from seizures and altered intersegmental coordination during locomotion, thus indicating that a setpoint of network excitability is determined during the CP. Next, **compelling** experiments demonstrate that this goes along with increased excitation/inhibition ratios to single identified motoneurons and most importantly for excitability setpoint determination during the CP excitatory and inhibitory inputs are integrated such that the effect of CP hyperexcitation is rescued by the stimulation of endogenous inhibitory inputs to the motoneurons. This provides novel insight into how developing neural network excitability is tuned and how it can be entrained during the CP.

---

## [Referee Report · Reviewer #1 (Public Review)]

Activity has effects on the development of neural circuitry during almost any step of neuronal differentiation. In particular during specific time periods of circuit development, so called critical periods (CP), altered neural activity can induce permanent changes of neuronal and network excitability. In complex neural networks it is often difficult to pinpoint the specific network components that are permanently altered by activity, and it often remains unclear how activity is integrated during the CP to set mature network excitability. This study combines electrophysiology with pharmacological and optogenetic manipulation in the *Drosophila* genetic model system to pinpoint the neural substrate that is influenced by altered activity during a critical period (CP) of larval locomotor circuit development. Moreover, it is then tested whether and how different manipulations of synaptic input are integrated during the CP to tune network excitability.

Strengths: Based on previous work, during the CP network activity is increased by feeding the GABA-AR antagonist PTX. This results in permanent network activity changes as highly convincingly assayed by a prolonged recovery period following induced seizure and by altered intersegmental locomotor network coordination. This is then used to provide two important findings: First, compelling electro- and optophysiological as well as anatomical experiments track the site of network change down to the level of single neurons and pre- versus postsynaptic specializations. In short, increased activity during the CP increases both, the magnitude of excitatory and inhibitory synaptic transmission to the aCC motoneuron, but excitation is affected more strongly. This results in altered excitation inhibition ratios. Fine electrophysiology shows that excitatory synapse strengthening occurs postsynaptically. High quality anatomy shows that dendrite size and numbers of synaptic contacts remain unaltered. It is a major accomplishment to track the tuning of network excitability during the CP down to the physiology of specific synapses at identified neurons.

Second, additional experiments with single neuron resolution demonstrate that during the CP different forms of activity manipulation are integrated so that opposing manipulations can rescue altered setpoints. This provides novel insight into how developing neural network excitability is tuned, and it indicates that during the CP training can rescue the effects of hyperactivity.

Weaknesses: There are no major weaknesses to the findings presented, but the molecular cause that underlies increased motoneuron postsynaptic responsiveness as well as the mechanism that integrates different forms of activity during the CP remain unknown. However, the discussion addresses this point adequately.

---

## [Referee Report · Reviewer #2 (Public Review)]

SUMMARY: In this study, the authors use the tractable *Drosophila* embryonic/larval motor circuit to determine how manipulations to activity during a critical period (CP) modify the circuit in ways that persist into later developmental stages. Previously, this group demonstrated that manipulations to the aCC/MN-Ib neuron in embryonic stages enhance (or can rescue) susceptibility to seizures at later larval stages. Here, the authors demonstrate that following enhanced excitatory drive (by PTX feeding), the aCC neuron acquires increased sensitivity to cholinergic excitatory transmission, presumably due to increased postsynaptic receptor abundance and/or sensitivity, although this is not clarified. Although locomotion is not altered at later developmental larval stages, the authors suggest there is reduced "robustness" to induced seizures. The second part of the study then goes on to enhance inhibition during the CP in an attempt to counteract the enhanced excitation, and show that many aspects of the CP plasticity are rescued. The author conclude that "average" E/I activity is integrated during the CP to determine excitability of the mature locomotor network.

Overall, this study provides compelling mechanistic insight into how a final motor output neuron changes in response to enhanced excitatory drive during a CP to change functionality of the circuit at later mature developmental stages. The first part of this study is strong, clearly showing the changes in the aCC neuron that result from enhanced excitatory input. This includes very nice electrophysiology and imaging data that assess synaptic function and structure onto aCC neurons from pre-motor inputs resulting from PTX exposure during development. However, the later experiments in Figures 6 and 7 designed to counteract the CP plasticity are somewhat difficult to interpret. In particular, the specificity of the manipulations of the ch neuron intended to counteract the CP plasticity is unclear, given the complexities of how these changes impact excitability all neurons during development. It is clear that CP plasticity is largely rescued in later stages, but it is hard to know if downstream or secondary adaptations may be masking the PTX-induced plasticity normally observed. Nonetheless, this study provides an important advance in our understanding of what parameters change during CPs to calibrate network dynamics at later developmental stages.

---

## [Referee Report · Reviewer #3 (Public Review)]

Summary:

In Hunter, Coulson et al, the authors seek to expand our understanding of how neural activity during developmental critical periods might control the function of the nervous system later in life. To achieve increased excitation, the authors build on their previous results and apply picrotoxin 17-19 hours after egg-laying, which is a critical period of nervous system development. This early enhancement of excitation leads to multiple effects in third-instar larvae, including prolonged recovery from electroshock, increased synchronization of motor neuron networks, and increased AP firing frequency. Using optogenetics and whole-cell patch clamp electrophysiology, the authors elegantly show that picrotoxin-induced over-excitation leads to increased strength of excitatory inputs, and not loss of inhibitory inputs. To enhance inhibition, the authors chose an approach that involved stimulation of mechanosensory neurons; this counteracts picrotoxin-induced signs of increased excitation. This approach to enhancing inhibition requires further validation.

Strengths:

• The authors confirm their previous results and show that 17-19 hours after egg laying is a critical period of nervous system development.

• Using Ca2+/Sr2+ substitutions, the authors demonstrate that synaptic connections between A18a & aCC show increased mEPSP amplitudes. The authors show that this aCC input is what is driving enhanced excitation.

• The authors demonstrate that the effects of over-excitation attributed to picrotoxin exposure are generalizable and also occur in bss mutant flies.

Weaknesses:

• The authors build on their previous work and argue that the critical period (17-19h after egg-laying) is a uniquely sensitive period of development. Establishing the developmental window of the critical period is important for the present study. The present study would benefit from demonstrating that exposure to picrotoxin at L1 or L2 do not lead to changes in induced seizure at L3. This would further the authors hypothesis of the criticality of the 17-19h AEL period.

• The ch-related experiments require further controls and explanation. Regarding experiments in Fig 6, what is the effect of ch neuron stimulation alone on time lag and AP frequency? The authors report related pilot experiments have been performed; the present study would be strengthened with inclusion of these data.

---

## [Author Response]

The following is the authors’ response to the original reviews.

**Public Reviews:**

**Reviewer #1 (Public Review):**
Activity has effects on the development of neural circuitry during almost any step of differentiation. In particular during specific time periods of circuit development, so-called critical periods (CP), altered neural activity can induce permanent changes in network excitability. In complex neural networks, it is often difficult to pinpoint the specific network components that are permanently altered by activity, and it often remains unclear how activity is integrated during the CP to set mature network excitability. This study combines electrophysiology with pharmacological and optogenetic manipulation in the *Drosophila* genetic model system to pinpoint the neural substrate that is influenced by altered activity during a critical period (CP) of larval locomotor circuit development. Moreover, it is then tested whether and how different manipulations of synaptic input are integrated during the CP to tune network excitability.Strengths:Based on previous work, during the CP, network activity is increased by feeding the GABA-AR antagonist PTX. This results in permanent network activity changes, as highly convincingly assayed by a prolonged recovery period following induced seizure and by altered intersegmental locomotor network coordination. This is then used to provide two important findings: First, compelling electro- and optophysiological experiments track the site of network change down to the level of single neurons and pre- versus postsynaptic specializations. In short, increased activity during the CP increases both the magnitude of excitatory and inhibitory synaptic transmission to the aCC motoneuron, but excitation is affected more strongly. This results in altered excitation inhibition ratios. Fine electrophysiology shows that excitatory synapse strengthening occurs postsynaptically. High-quality anatomy shows that dendrite size and numbers of synaptic contacts remain unaltered. It is a major accomplishment to track the tuning of network excitability during the CP down to the physiology of specific synapses to identified neurons.Second, additional experiments with single neuron resolution demonstrate that during the CP different forms of activity manipulation are integrated so that opposing manipulations can rescue altered setpoints. This provides novel insight into how developing neural network excitability is tuned, and it indicates that during the CP, training can rescue the effects of hyperactivity.Weaknesses:There are no major weaknesses to the findings presented, but the molecular cause that underlies increased motoneuron postsynaptic responsiveness as well as the mechanism that integrates different forms of activity during the CP remain unknown. It is clear that addressing these experimentally is beyond the scope of this study, but some discussion about different candidates would be helpful.

We discuss likely mechanisms that underpin the increase in postsynaptic responsiveness below (Reviewer #1 (Recommendations For The Authors):, point 2). To address possible mechanisms that integrate different forms of activity we now include a new paragraph in the discussion.

**Reviewer #2 (Public Review):**
Summary:In this study, the authors use the tractable *Drosophila* embryonic/larval motor circuit to determine how manipulations of activity during a critical period (CP) modify the circuit in ways that persist into later developmental stages. Previously, this group demonstrated that manipulations to the aCC/MN-Ib neuron in embryonic stages enhance (or can rescue) susceptibility to seizures at later larval stages. Here, the authors demonstrate that following enhanced excitatory drive (by PTX feeding), the aCC neuron acquires increased sensitivity to cholinergic excitatory transmission, presumably due to increased postsynaptic receptor abundance and/or sensitivity, although this is not clarified. Although locomotion is not altered at later developmental larval stages, the authors suggest there is reduced "robustness" to induced seizures. The second part of the study then goes on to enhance inhibition during the CP in an attempt to counteract the enhanced excitation, and show that many aspects of the CP plasticity are rescued. The authors conclude that "average" E/I activity is integrated during the CP to determine the excitability of the mature locomotor network.Overall, this study provides compelling mechanistic insight into how a final motor output neuron changes in response to enhanced excitatory drive during a CP to change the functionality of the circuit at later mature developmental stages. The first part of this study is strong, clearly showing the changes in the aCC neuron that result from enhanced excitatory input. This includes very nice electrophysiology and imaging data that assess synaptic function and structure onto aCC neurons from pre-motor inputs resulting from PTX exposure during development. However, the later experiments in Figures 6 and 7 designed to counteract the CP plasticity are somewhat difficult to interpret. In particular, the specificity of the manipulations of the ch neuron intended to counteract the CP plasticity is unclear, given the complexities of how these changes impact the excitability of all neurons during development. It is clear that CP plasticity is largely rescued in later stages, but it is hard to know if downstream or secondary adaptations may be masking the PTX-induced plasticity normally observed. Nonetheless, this study provides an important advance in our understanding of what parameters change during CPs to calibrate network dynamics at later developmental stages.
**Reviewer #3 (Public Review):**
Summary:In Hunter, Coulson et al, the authors seek to expand our understanding of how neural activity during developmental critical periods might control the function of the nervous system later in life. To achieve increased excitation, the authors build on their previous results and apply picrotoxin 17-19 hours after egg-laying, which is a critical period of nervous system development. This early enhancement of excitation leads to multiple effects in third-instar larvae, including prolonged recovery from electroshock, increased synchronization of motor neuron networks, and increased AP firing frequency. Using optogenetics and whole-cell patch clamp electrophysiology, the authors elegantly show that picrotoxin-induced over-excitation leads to increased strength of excitatory inputs and not loss of inhibitory inputs. To enhance inhibition, the authors chose an approach that involved the stimulation of mechanosensory neurons; this counteracts picrotoxin-induced signs of increased excitation. This approach to enhancing inhibition requires further control experiments and validation.Strengths:• The authors confirm their previous results and show that 17-19 hours after egg laying is a critical period of nervous system development.• Using Ca2+/Sr2+ substitutions, the authors demonstrate that synaptic connections between A18a aCC show increased mEPSP amplitudes. The authors show that this aCC input is what is driving enhanced excitation.• The authors demonstrate that the effects of over-excitation attributed to picrotoxin exposure are generalizable and also occur in bss mutant flies.Weaknesses:• The authors build on their previous work and argue that the critical period (17-19h after egg-laying) is a uniquely sensitive period of development. Have the authors already demonstrated that exposure to picrotoxin at L1 or L2 (and even early L3 if experimentally possible) does not lead to changes in induced seizure at L3? This would further the authors' hypothesis of the uniqueness of the 17-19h AEL period. If this has already been established in prior publications, then this needs to be further explained. I do note in Gaicehllo and Baines (2015) that Fig 2E shows the identification of the 17-19h window.

This is a pertinent comment. We now have evidence that activity manipulation (in this instance by increasing temperature, which recapitulates the effect of PTX) is not effective at larval stages (L1 to L3) but remains effective between 17-19hrs AEL. This observation forms part of a separate study where we explore the role of circadian activity on embryonic and larval neuronal development. We include a brief statement to address this comment in the revision (first paragraph of Results).

• Regarding experiments in Fig 2, authors only report changes in AP firing frequency. Can the authors also report other metrics of excitability, including measures of intrinsic excitability with and without picrotoxin exposure (including RMP, Rm)? Was a different amount of current injection needed to evoke stable 5-10 Hz firing with and without picrotoxin? In the representative figure (Fig. 2A), it appears that the baseline firing frequencies are different prior to optogenetic stimulation.

No differences in RM, Rin or capacitance were observed due to PTX. This is now included in the revision along with an explanation that different levels of current injection were used to measure effects to excitatory vs inhibitory synaptic drive. We did not specifically monitor the amount of current required to maintain stable firing.

• The ch-related experiments require further controls and explanation. Regarding experiments in Fig 6, what is the effect of ch neuron stimulation alone on time lag and AP frequency? Can the authors further clarify what is known about connections between aCC and ch neurons? It is difficult for this reviewer to conceptualize how enhancing ch-mediated inhibition would worsen seizures. While the cited study (Carreira-Rosario et al 2021) convincingly shows that inhibition of mechanosensory input leads to excessive spontaneous network activity, has it been shown that the converse - stimulation of ch neurons - indeed enhances network inhibition?• The interpretation of ch-related experiments is further complicated by the explanation in the Discussion that ch neuron stimulation depolarizes aCC neurons; this seems to undercut the authors' previous explanation that the increased E:I ratio is corrected by enhanced inhibition from ch neurons. The idea that ch neurons are placing neurons in a depolarized refractory state is not substantiated by data in the paper or citations.

To respond to these two points combined: The reviewer is correct in stating that additional experiments will be required to fully understand mechanism. We believe that cholinergic (excitatory) chordotonal input to aCC may be an important component for setting the rhythm of the locomotor CPG. Indeed, it may be that CPG rhythm is a key factor during the CP. Our observations suggest optogenetic stimulation of Ch neurons alone is sufficient to induce large, ~400-, currents that resemble endogenous spontaneous rhythmic currents (SRCs) associated with CPG activity. SRCs occur with a characteristic frequency of ~1Hz, and we have some unpublished data that suggests it is possible to change this frequency using ch stimulation. This data therefore unifies prior work (Carreira-Rosario et al., 2021 description of a brake) with our own (observation that ch depolarize aCC). However, we do not include this speculation in the Discussion because the experiments we have conducted were pilots. They may be expanded upon and included in future work.

• In the Discussion, the authors suggest that enhanced proprioception leading to seizures is reminiscent of neurological conditions. This seems to be an oversimplification. Connecting abnormal proprioception to seizures is quite different from connecting abnormal proprioception to disorders of coordination. This should be revised.

Because this is peripheral to our main study, we have deleted this from the revision.

**Reviewer #1 (Recommendations For The Authors):**
1. Although the authors have to be commended for the scrutiny with which they pinpoint a site of circuit change, it cannot be excluded that other parts of the circuit also undergo adjustments in response to activity manipulation during the CP, e.g. the membrane properties of the interneurons. This is not a problem but should be discussed.

We agree with this comment and have added the following text to the discussion……’However, we recognise that other parts of the locomotor network may also undergo change due to CP manipulation. The advantage of this system is that most of these elements are now open to specific manipulation through cell-specific genetic drivers’. (Discussion paragraph 3)

2. It is surprising that there is no discussion of the potential molecular cause for the observed increases in postsynaptic responses to SV release from cholinergic neurons. Given that there are no differences in postsynaptic structure, puncta number etc., the subunit composition of the nAChR seems an obvious guess. What is known about the nAChRs subunit composition on aCC, and when during development do the receptors/different subunits become expressed? A paragraph in the discussion on this issue would be highly relevant to the manuscript.

Our own work (unpublished) together with a recent paper from the Littleton lab suggests that aCC expresses the majority, if not all, of the 7 alpha and 3 beta subunits that compromise nAChRs. The situation is further complicated by the fact that these receptors are pentameric and are composed of various subunits – the composition significantly altering channel kinetics. Less is known about expression timelines for each receptor subunit, and certainly not in aCC. We already include the following sentence in the results text……’ A change in the frequency of mini excitatory postsynaptic potentials (mEPSPs, a.k.a. minis) would suggest the adaptation is primarily presynaptic (e.g. increased probability of release), whilst a change in distribution and/or amplitude of minis is more consistent with a mechanism acting postsynaptically (e.g. increased or altered receptor subunits).’ Given that we know next to nothing about the nAChR subunit composition in aCC and how this might change due to CP manipulation, we feel it better not to speculate further. To help the reader, we include the following sentence in the discussion……’The precise mechanism contributing to increased mini amplitude remains to be determined, but a plausible scenario may involve change in cholinergic subunit composition.’ (Discussion paragraph 3)

3. It would be important to provide the p-values for Figures 1B and C, especially because it seems that the inhibition also becomes stronger upon PTX treatment during the CP. There is no statistical testing mentioned, was no test done or was it not significant? It is agreed that the effect size is clearly stronger for the increased excitation than for the increased inhibition, but looking at the data suggests that the effect on excitation is not much more significant than the effect on inhibition.

The reviewer is referring to Fig 2B&C. P values have been added to both main text and to the figure legend.

4. Associated with the point above, in the discussion line 407 and below the authors come back to this point and reason that it is surprising that increased excitation is not compensated for by homeostatic mechanisms. It is concluded that homeostatic compensation brings the system back to a setpoint that is defined during the critical period, but the setpoint is set higher in this case. However, an alternative explanation is that GABA administration during the critical period causes the excitation set point to be too high, but this is then partially counteracted in a homeostatic manner by increasing inhibition. If the p-values in Figures 2B and C are rather similar, this might even be the favorable interpretation.

We believe the reviewer means ‘PTX administration’ and not GABA. This is an interesting idea and one we had not really considered. We address this comment by adding the following text………. ‘Alternatively, whilst the increased inhibition we observe is not statistically significant (p = 0.15), it is close and has a medium effect size (Cohen’s d = 0.78), and thus may be indicative of an attempt by the locomotor network to rebalance activity back towards a genetically pre-determined level. In this regard, it may just not have sufficient range to be able to counter the increase in excitation due to CP manipulation.’ (Discussion paragraph 5)

5. To asses the magnitudes of A18a-mediated excitation and A31k-mediated inhibition to aCC, changes in aCC firing frequency were measured. For this aCC was injected with current to fire at all. However, the current injections were chosen to cause firing at 5-10 Hz. During a crawling burst, aCC fires well above 100Hz (Kadas et al., 2017). Are the effects also visible at such firing frequencies, or at least across different firing frequencies? I am not asking for additional experiments, but maybe the data are there and can be referred to?

Spiking in aCC occurs as burst firing, evoked by cholinergic synaptic drive, that lasts for ~300ms and achieving firing frequencies of between 50-100Hz (Kadas et al., 2017 and our own unpublished data). We did not test for effects to excitation or inhibition at these higher frequencies. We now make this explicit in the discussion by adding the following sentence……’The firing frequencies that we imposed (1-10Hz) are also lower than seen during fictive locomotion (Kadas et al., 2017), which shows burst firing lasting for ~300 ms and achieving spike frequencies of up to 100Hz.’ (Discussion paragraph 3)

6. In Figure 3B some minis are demarked by green arrows and others are not. Were the non-marked ones not included in the analysis, and what were the criteria to mark some and others not? This is particularly important because the cumulative distribution of minis is analyzed in Figure 3D, and this depends crucially on what qualifies as mini and what does not.

All mini’s are marked by green arrows. The events not marked are not mini’s. *Drosophila* neurons are small and have an unfavourable dendritic structure for recording minis. Thus, we carefully analyse traces by eye taking only events that show very rapid rise times and slower, exponential decay (the typical mini shape). There are, however, other events which are most likely single/multiple channel openings, which due to filtering are rounded. We now include this same trace, greatly expanded, as Fig S1D to show how we identified minis from non-minis.

7. The asynchronous release experiment under Sr2+ seems an elegant way to analyze minis upon optogenetic stimulation of an identified presynaptic cholinergic neuron. I suggest being a little more conservative with the term asynchronous release (or replacing it), which is usually the release of many single vesicles that follow AP-mediated synaptic transmission and has nicely been demonstrated at the *Drosophila* NMJ (Besse et al., 2007). Also, please show the trace in Figure S2A under Sr2+ at a higher pA magnification, it is really hard to see the minis there.

We have adopted a previously published technique that, in our view, correctly uses the term ‘asynchronous release’. This is not to say that all asynchronous release occurs via the same mechanism. Indeed, the papers that report the technique we use predate Besse 2007. We also expand the trace in Fig S1A (not S2A as wrongly indicated).

**Reviewer #2 (Recommendations For The Authors):**
1. Can the authors explain what they think is the parameter of "activity" being measured in the locomotor circuit (mainly aCC) during the CP? Is the aCC neuron simply summing (perhaps through a proxy like Ca2+) total excitation/inhibition over time during the CP?

Reviewer #1 also requests that we discuss how activity is ‘measured’ and thus we now include a dedicated paragraph in the discussion to address this concern. Whether aCC sums ‘average’ activity or perhaps is influenced by activity extremes remains uncertain. Our data is consistent with the former but further work is required to validate our conclusion. This work will be published in due course.

Related to understanding this concept, could the authors' silence activity (using Kir2.1, TNT, or BoNT) from each of the monosynaptic premotor inputs in otherwise wildtype and following PTX exposure to determine how the circuit responds when each of the monosynaptic inputs are silenced? This might inform the role they play in instructing how activity is measured over time during the CP.

This is an excellent suggestion and, indeed, we have planned such experiments. Silencing specific neurons, whilst manipulating the CP, may well result in more significant network instability due to the setting of multiple (and physiologically inappropriate) homeostatic set points. Such studies go beyond the scope of the present study and thus we prefer not to speculate at this early stage, but to wait for experimental data.

On a related note, the authors focus on just 2 premotor inputs, presumably due to the availability of specific drivers. But do the authors know how many other inputs (other ACh, Gaba, and glutamate) onto aCC there are, and to what extent do the authors think these are changed in similar or distinct ways? Is it implied that all neurons are similarly altered by the manipulations?

The connectome details the number and types of neurons that directly contact the aCC motoneuron (Zarin et al., 2019). In terms of cholinergic excitors, the results present in Figure 3 suggest that most (all?) inputs are strengthened following embryonic PTX exposure. However, to conclude this would be highly speculative and thus we refrain from doing so in the manuscript. As other single-neuron driver lines become available, such expts will hopefully be possible.

2. If PTX treatment does indeed increase CPG synchronicity, shouldn't there be a readout of this effect on larval locomotion? While the speed of locomotion wasn't significantly impacted, perhaps another parameter was altered.

It is quite possible that other aspects of locomotion are being altered (turning, rearing, etc), but we have not analysed for these more subtle behaviours. Indeed, although not statistically significant, there is a modest reduction in average velocity in larvae derived from PTX-exposed embryos. We see similar reductions in characterised seizure mutants which also show increased synchronicity (Streit et al., 2016).

3. In Figure 2 and elsewhere, what is the baseline level of AP firing rate in each aCC neuron, before optogenetic stimulation? Is this informative about how PTX exposure alters excitability to begin with, perhaps by changing intrinsic excitability.

We now include this data in the relevant results section. Interestingly, following exposure to PTX, basal firing was significantly increased in A18a (excitatory premotor) but not in A31k (inhibitory premotor). This reflects our experiment in which we conclude that excitatory drive to aCC is increased relative to inhibitory synaptic drive. Thus, this measure seemingly validates our conclusion that E:I balance has been altered following activity-manipulation during the CP.

4. Figure 3: The apparent increase in mini amplitude is very small (4.1 vs 4.5 pA); is this physiologically meaningful? Although the authors say the decrease in mini freq is not significant in Fig. 3B after PTX, it does appear rather large, a 40% reduction (5 vs 3 Hz).

We must be guided by statistics in drawing conclusions, but the reader can interpret our data as they wish. Minis measure quantal release and thus to appreciate how small change can, when combined over the many receptors present, influence cell physiology, one needs to compare spiking activity. We show in Fig 2 that such change is sufficient to increase the excitatory synaptic drive provided by the A18a neuron. The seemingly larger reduction in mini frequency is intriguing and may reflect additional change, but without further experiments we cannot draw firm conclusions.

5. The clever vibration assay is a good one to induce the activation of mechanosensory neurons, but the specificity of the changes induced by this is difficult to ascertain. One possibility would be to silence the output of the ch neurons (by expression to tetanus or botulinum toxin) and still put the larvae through the same vibration during the CP to see if the rescue is lost.

We agree that further experiments are required to fully understand underlying mechanism(s). However, we will not be able to complete such follow-on expts in a timely manner and thus, these must wait and form the basis of future studies.

Minor points1. Typos - there are numerous areas where it seems a comma is used inappropriately (e.g. lines 28, 69, 77, 104, 348, 365, etc). Suggest line editing the final "version of record".

Checked and corrected.

2. It would be of benefit to show the genotypes of the larvae in the various experimental manipulations in the relevant figure legends. This reviewer could not follow exactly how each experiment was done as it was not always clear which driver was being used to express which transgene in what genetic background.

Done

**Reviewer #3 (Recommendations For The Authors):**
• Please provide sample videos of electroshock-induced seizures (e.g. Fig 1B). Is it clear that the period of immobility after electroshock is a seizure (perhaps defined as hyperactivity originating from the brain)? I acknowledge the Baines group is quite skilled in this technique and perhaps there is a straightforward answer or citation to include.

We refer the reader to Marley and Baines 2011 which contains videos of seizure activity (first paragraph of Results).

• Seizures are generated in the brain and travel to the periphery. Do the authors think it is possible that the peripheral manipulations in this manuscript might be controlling the behavioral readout of seizures without affecting hypersynchronous activity in the brain?

We include the following statement (in methods) to provide our best understanding for how peripheral electroshock induces seizure………. ‘Strong peripheral stimulation likely causes excessive and synchronous synaptic excitation within the CNS resulting in seizure. However, the precise mechanism of this effect remains to be determined.’ Moreover, we feel it unlikely that manipulation of Ch neurons, by vibration, would suppress the effects we observe via peripheral mechanisms. Indeed, the Ch manipulation is limited to the embryonic CP, whilst our seizure assays are recorded many days later at L3.

• How might enhancement of inhibition lead to worsened seizures? Is the enhancement of ch-related inhibition selectively affecting inhibitory circuits, thereby leading to a net increase in excitation?

This is a difficult point to respond to at present. Enhanced inhibition per se might similarly disturb the encoding of an appropriate homeostatic setpoint(s) thus leaving a network open to being destabilized by a strong stimulus. Indeed, we have previously shown that increased inhibition during the CP results in the same effect (seizure) as increasing excitation (Giachello and Baines, 2015). Thus, presuming activation of Ch neurons during the CP translates to increased inhibition, then worsened seizure behaviour is a predictable effect. How this is achieved remains unknown and we prefer not to speculate here.